# The spatio-temporal patterns and formation mechanisms of cholera epidemics in Hubei Province, China from 1949 to 2020

Zhiyu Chen, Shengsheng Gong, Tao Zhang *

Hubei Key Laboratory for Geographical Process Analysis and Simulation, Central China Normal University,Wuhan, Hubei, China

* ztccnu@ccnu.edu.cn

## Abstract

### Background

The global epidemics of cholera, a virulent enteric infection, pose a serious threat to public health and socio-economics. The disease's rapid spread and high mortality have led to heavy casualties, along with disruptions to production, rising healthcare burdens and impaired economic exchanges.

### Methods

Based on cholera historical and environmental data, Mann-Kendall test, wavelet analysis, hotspot analysis, epidemics center of gravity, and structural equation modeling were employed to investigate the spatial and temporal distribution pattern and formation mechanism of cholera epidemics in Hubei Province during the period of 1949–2020.

### Results

Based on existing historical monitoring records. Temporally, high incidence occurred in the 1980s and 1990s, with summer and autumn being the dominant seasons. Three fluctuation cycles were identified: 29 years, 19 years, and 8 years. Spatially, 71 counties and districts were cumulatively affected, with Wuhan and Xianning serving as the primary hotspots. Cholera epidemics were distributed along rivers and lakes, and their centers of gravity shifted southward and westward over time. Mechanistically, cholera epidemics were the result of the combined effects of natural, disaster, and human factors. Population density and summer temperatures were the direct factors driving the spread of cholera epidemics, while river networks formed the basic environmental background that facilitated transmission. Natural environmental factors such as drought and flood disasters, elevation, and precipitation, as well as human environmental factors including road networks and economic conditions, could not

**Data availability statement:** All data are in the manuscript and/or Supporting information files.

**Funding:** This work was supported by the National Natural Science Foundation of China (42471285 to TZ; 42371265 to SG; 41801141 to TZ), National Social Science Foundation (21VJXT01 to SG; 24FZSB0795 to TZ), Major Program of Philosophy and Social Science Research in Hubei Higher Education Institutions (23ZD248 to TZ), 2024 Open Fund Project of Hubei Key Laboratory of Geographic Process Analysis and Simulation (ZDSYS202403 to TZ) and 2025 Fundamental Research Funds for the Central Universities for Interdisciplinary Platform Construction for Analysis and Simulation of Geographical Processes in the Middle Reaches of the Yangtze River (CCNU25JCPT006 to TZ). The funders have no role in the conceptualization, design, data collection, analysis, decision to publish, or preparation of the manuscript.

**Competing interests:** The authors have declared that no competing interests exist.

only directly drive the spread of epidemics but also regulate epidemics development through indirect pathways. The coupling effects of various factors across different spatiotemporal scales jointly shaped the unique spatiotemporal distribution and evolutionary characteristics of cholera epidemics.

## Conclusion/Significance

This study helped to reveal the spatiotemporal patterns and formation mechanisms of regional cholera epidemics, filled the gap in cholera research in Hubei Province, and provided a reference for cholera prevention and control in areas with dense rivers and lakes.

## Author summary

Cholera remains a major threat to public health, especially in regions with dense water networks. As a "province of a thousand lakes" in China, Hubei Province has a long history of cholera epidemics, but its long-term spatiotemporal patterns and driving forces were not fully understood. We studied cholera cases in Hubei Province from 1949 to 2020, combining historical records and environmental data with analytical tools like spatial analysis and statistical modeling. Based on existing historical monitoring records, we found cholera epidemics were most severe in the 1980s–1990s, peaked in summer and autumn, and had three cycles (29, 19, 8 years). Spatially, it clustered in eastern Hubei Province along rivers/lakes, with its center shifting southwest over time. Population density and summer temperatures were the direct factors driving the outbreak of cholera epidemics, while river networks constituted the basic environmental background that facilitated epidemics transmission. Natural environmental factors such as elevation and precipitation, human environmental factors including road networks and economic conditions, as well as drought and flood disasters, could not only directly drive the spread of epidemics but also regulate the development of epidemics through indirect pathways. Our findings filled gaps in Hubei's cholera research and provided targeted containment and prevention clues for water-rich regions globally, thereby helping to balance environmental protection and disease control.

## Introduction

Cholera is one of only two Class A statutory infectious diseases in China (the highest hazard level in China's infectious disease classification system), a virulent intestinal infectious disease caused by *Vibrio cholerae* [1]. In terms of spatiotemporal patterns, cholera was likely introduced into China by sea around the Jiaqing-Daoguang period (circa 1796–1850) of the Qing Dynasty. It first landed in Guangzhou and then gradually spread to coastal areas and inland regions [2]. After 1949, with the advancement

of prevention and control measures, the national cholera incidence rate dropped significantly. However, there were still three epidemic peaks in the 1960s, 1980s, and 1990s, predominantly caused by the *El Tor biotype of Vibrio cholerae* (a subtype of serogroup O1, known for its strong environmental survival and transmission potential). It was not until 1993 that the first local outbreak of *O139 cholera* (cholera caused by the O139 serogroup of *Vibrio cholerae*) occurred, and since then, *O139 cholera* epidemics have continued unabated [3]. As the "Province of a Thousand Lakes" and a transportation hub in the middle reaches of the Yangtze River, Hubei's dense network of rivers and lakes provides a unique environmental substrate for cholera transmission. Combined with the impact of major human activities such as the Three Gorges Project on the hydrological environment, and the frequent population movements resulting from its role as a core transportation node, the region serves as a prototypical area for studying the spatiotemporal dynamics of cholera. Historically, Hubei Province had been hit by severe cholera epidemics on multiple occasions: in the first year of Xuantong (1909), those who died of cholera in Hankou were "omnipresent" [4]. In the 21st year of the Republic of China (1932), cholera occurred in 12 cities in Hubei Province, with over 1,200 people dying from it, and the mortality rate reached 43.5% [5]. Since 1949, especially since the year 2000, the incidence rate of intestinal infectious diseases such as cholera had shown a year-by-year downward trend [6]. The epidemics are mainly characterized by sporadic cases and small-scale outbreaks caused by group dining events [7,8]. The incidence rises significantly in summer and autumn [9], and high-incidence areas are mostly concentrated in counties along rivers, such as the Jianghan Plain [10]. In terms of the formation mechanism, climate warming can to a certain extent lead to the transmission and spread of cholera [11]. As a waterborne disease, cholera is transmitted through contaminated well water and river water [12]. Hilly areas, mountainous areas, and plain areas are all high-incidence topographical regions [13,14]. The development of transportation has provided conditions for the transmission of cholera [15]. Cholera often follows disasters. Among them, wars cause frequent outbreaks of cholera by destroying infrastructure, creating population displacement and resource scarcity [16]; floods significantly increase the risk of cholera epidemics by contaminating water sources, damaging sanitation systems, and exacerbating population vulnerability [17]. For example, the catastrophic 1998 floods in the Yangtze River basin heavily impacted Hubei—one of the most severely affected regions—and provided a valuable natural experiment for examining how extreme climate events trigger cholera epidemics.

However, existing studies still have some limitations: In terms of spatiotemporal patterns, analyses have mainly focused on the spatiotemporal patterns of cholera in some regions [18–22]. There is still a lack of geographical studies with long-term and high spatiotemporal resolution on regional cholera epidemics. In terms of formation mechanisms, there is still a lack of comprehensive quantitative studies on the formation mechanisms of regional cholera epidemics, and there is also insufficient identification of major influencing factors and analysis of their action mechanisms [18,23]. In terms of cholera research in Hubei Province, there is a lack of specialized studies on cholera epidemics in this region, which is densely covered with rivers and lakes and has a severe history of epidemics. Therefore, to address these research gaps, this study aims to systematically answer three core scientific questions: (1) The long-term trends, cyclical fluctuations, and seasonal characteristics of cholera epidemics in Hubei Province from 1949 to 2020; (2) Its overall distribution patterns and spatial distribution changes; (3) The formation mechanisms of these spatiotemporal patterns shaped by multidimensional factors including natural conditions, disaster events, and socioeconomic factors. To this end, this study integrates 72 years of county-level cholera epidemic records from Hubei Province spanning 1949–2020 with multidimensional environmental data. Employing a comprehensive methodology including time series analysis, spatial statistics, and structural equation modeling, it aims to reveal the complex mechanisms underlying regional cholera epidemics through a long-term, high-spatial-resolution geographic analytical framework. It aims to provide scientific insights for cholera prevention and control in river- and lake-rich regions, particularly for other areas globally facing similar environmental conditions (e.g., monsoon climate, dense river networks) and socioeconomic pressures (e.g., rapid urbanization, high population mobility).

## Study area, data and methods

### Study area

Hubei Province has a unique geographical position, located in the middle reaches of the Yangtze River and north of Dongting Lake, with the terrain sloping gradually from northwest to southeast, surrounded by mountains in the east, west and north directions, while the central area is relatively flat. The terrain is complex and varied, with undulating hills, interspersed basins and lakes. The Yangtze River flows from the west to the east and runs through the entire province, which is rich in rivers and lakes and is known as the "Province of a Thousand Lakes". The climate is mild and humid, with mild winters and cool summers. The annual average temperature ranges between 15°C and 17°C. January (winter) averages 3–4°C, while July (summer) averages 27–29°C. Precipitation is abundant, with annual rainfall totaling 800–1600 millimeters, 40–50% of which falls during the summer months (June-August), characteristic of a typical subtropical monsoon climate. Hubei Province is located in the center of central China, and is an important transportation node connecting north and south and crossing east and west.

This study takes the administrative divisions of Hubei Province in 2020 as the study area, the county level as the basic spatial unit(103 counties in total), and the time range from 1949 to 2020 (Fig 1).

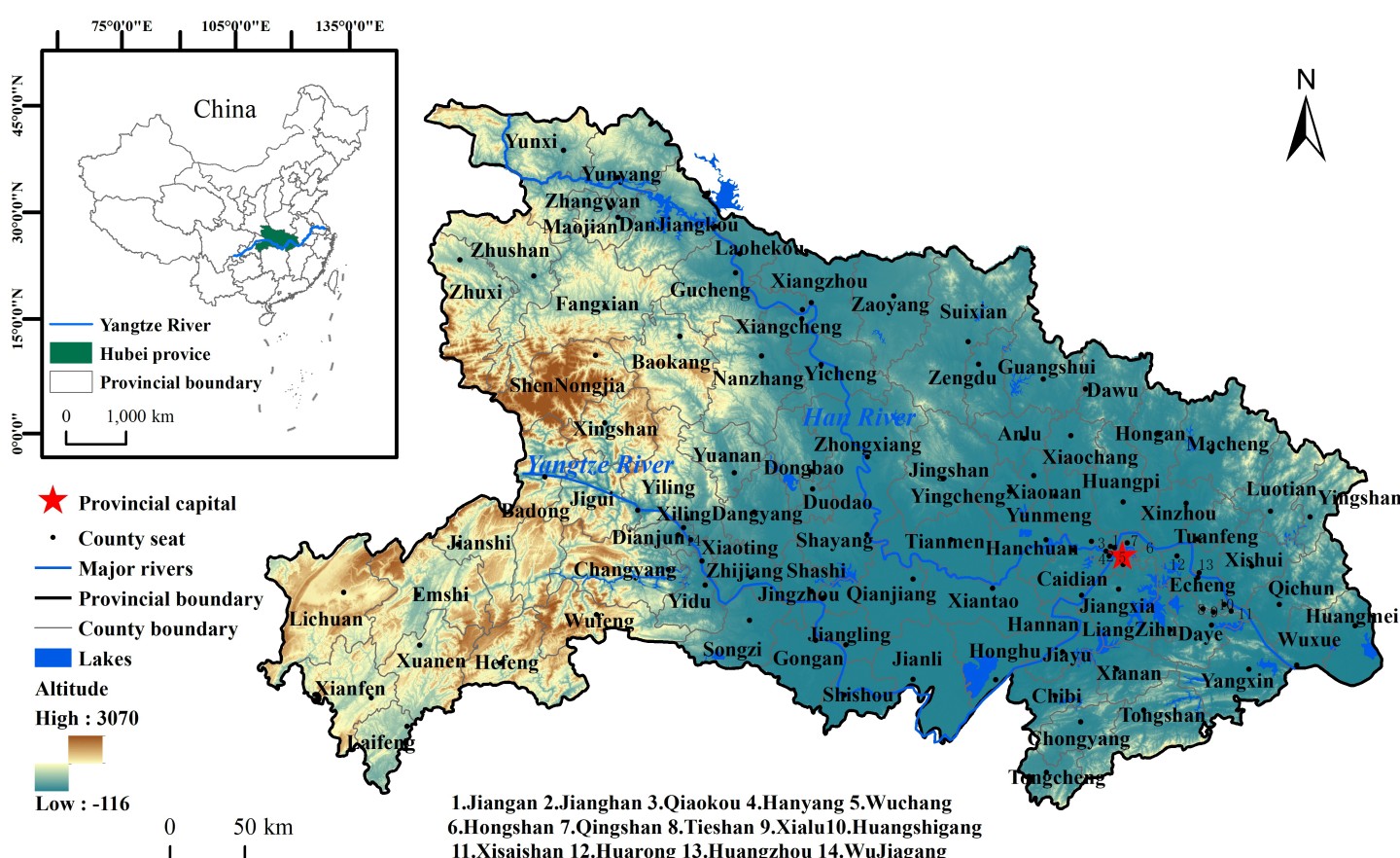

**Fig 1. Location of Hubei Province in China and Its Administrative Divisions.** Note: The basemap came from United States Geological Survey (https://apps.nationalmap.gov/services/), the map boundary has not been changed. Cartographic software: ArcGIS.

## Data source

[1] Cholera epidemics data. An epidemic refers to the prevalence of an infectious disease within a specific time and spatial scope. In this study, our subject is cholera, and we measure the intensity of cholera epidemics using the number of cholera cases as an indicator. Accordingly, we systematically reviewed all county annals, health annals, health yearbooks, and general yearbooks of Hubei Province from 1949 to 2020. Local documents with records of cholera epidemics were collated and included in S1 Table. On this basis, 166 pieces of data on cholera epidemics in Hubei Province were extracted, organized by county as the basic unit, and the number of cholera cases in each county per year was counted as the basic data for analysis in this paper. To ensure the spatial comparability of data across different periods, the study adopted the "spatial merging method for historical data": for county-level units that were abolished or merged between 1949 and 2020, their historical epidemics data were merged based on the 2020 county-level administrative divisions by comparing the changes in Hubei's administrative divisions during 1949–2020. Specific merging rules: For counties that were split—i.e., where a historical county-level unit was later divided into multiple 2020 county-level units—the number of cases was allocated according to the population proportion of the affected area in the year the epidemic occurred. Specifically, the most recent population census data for that epidemic year was obtained first. Then, the case numbers were weighted and allocated based on the proportion of the population in each post-split county relative to the total population of the original county. For example, if a county had a total population of 1 million in the epidemic year and was later split into County A (600,000 people) and County B (400,000 people), then 60% of the cholera cases reported for the original county would be assigned to County A and 40% to County B. For counties that underwent only name changes or jurisdictional adjustments with essentially unchanged boundaries, their historical data were directly matched to the 2020 county-level unit with the same name or covering the same geographical area. This method uses population as the allocation weight, based on the epidemiological characteristic that cholera transmission is closely related to population density and the frequency of human contact. All key time nodes of administrative division adjustments are detailed in S2 Table to ensure data traceability.

[2] Influencing factor data. The spread of an epidemic is mainly influenced by factors such as the source of infection, the route of transmission, and the susceptible population [24]. The reproduction and survival of *Vibrio cholerae* are highly dependent on environmental temperature, with its optimal growth temperature range being 16–42°C [25]. Ideally, water temperature is the most direct environmental factor driving the reproduction of *Vibrio cholerae*. However, given the long time span (72 years) and wide spatial scope (103 county-level units) of this study, it was not feasible to obtain consistent and continuous historical data on water temperature or aquatic Vibrio concentration. Summer air temperature is a key driver of surface water temperature, and there is a significant positive correlation between the two [26]. Meanwhile, cholera epidemics exhibit a distinct summer-autumn peak [27]. This peak shows a typical "time-lag effect" relative to summer high temperatures: high temperatures in summer promote the reproduction of Vibrio, leading to a peak in environmental bacterial load, which is then transmitted through water or food media, ultimately resulting in a case peak in late summer and early autumn (August-October). Therefore, this study selected the annual mean summer temperature as an important influencing factor of cholera infection sources. As a classic waterborne disease, cholera is primarily transmitted through contaminated water bodies [12]. Consequently, the density and distribution patterns of river networks form the fundamental geographic vehicle and spatial context for cholera transmission. Therefore, the density of river network is selected as an important factor influencing the spread of cholera in this study. The spread of epidemics is generally density-dependent, and cholera is no exception; the more densely populated an area is, the more susceptible people are [12], exposure to cholera is also simpler, therefore, population density is chosen as an important factor affecting cholera susceptibility in this study. Selection of annual mean summer temperature, river network density and population density as proxies for sources of infection, routes of transmission and susceptible populations, respectively, as direct influences and endogenous variables for cholera [27]. In addition, human factors

such as annual mean gross domestic product, nighttime lighting, and road network density, disaster factors such as floods and droughts, and natural factors such as mean elevation, annual mean precipitation, and annual mean hours of sunshine are selected as indirect influences and exogenous variables of the cholera epidemics to form a theoretical framework of the formation mechanism of the cholera epidemics as shown in Fig 2.

Annual mean summer temperature and precipitation data were obtained from the platform of the National Tibetan Plateau Scientific Data Center of China, based on the extraction and calculation of month-by-month raster data with 1-km resolution from 1949 to 2020 [28,29]. The mean elevation data were extracted from 30m resolution SRTMDEM data in Hubei Province; the river network data were obtained from Open Street Map; the sunshine data were generated from the high-resolution surface solar radiation dataset processed by the regional fusion of sunshine hours in China [30]. Flood and drought disaster data were from *China Meteorological Disaster Dictionary - Hubei Volume* [31]. Population data were mainly from the first to seventh censuses of China [32–38]. Road network density data is derived from the study entitled "Roads, Railroads and Decentralization of Chinese Cities" in Harvard Dataverse. Road network data for six years (1962, 1980, 1990, 2000, 2010, and 2020) in Hubei Province were extracted from it, and the average road network density was calculated [39]. The nighttime lighting data were obtained from the China "DMSP-OLS" 1km nighttime lighting remote sensing dataset of the National Geosystems Science Data Center [40]. Gross Domestic Product (GDP) data were from China County Statistical Yearbook and *Hubei Statistical Yearbook 1991* [41] et al. The selection of the aforementioned variables strictly adheres to the theoretical framework of "source of infection-mode of transmission-susceptible population" illustrated in Fig 2. It aims to characterize the core components of this classic epidemiological chain through quantifiable proxy variables [27].

The structural equation model in this study aims to identify the steady-state drivers shaping the long-term spatial patterns of cholera. To this end, we aggregated the data along the temporal dimension. Specifically: for each variable across all counties, we calculated a representative value spanning the entire study period from 1949 to 2020. For cholera case counts, we used cumulative total cases; for continuous variables like temperature and precipitation, we employed multi-year averages; for flood and drought disasters, we utilized cumulative occurrence years. Thus, the final data structure input to the model is a cross-sectional dataset comprising 103 observations, each representing a county's aggregated characteristics over 72 years. All variables involved in this study—including their definitions, units, time ranges, processing methods, and roles within the structural equation model—are summarized in Table 1.

Dataset on Cholera Epidemics and Environmental Factors in Hubei Province, China, 1949–2020 is deposited at https://zenodo.org/records/18008359.

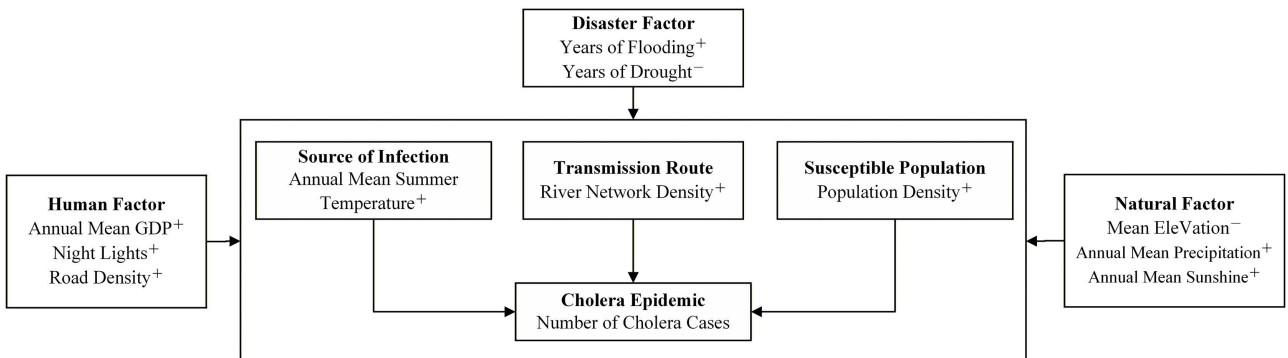

**Fig 2. Theoretical framework diagram of the formation mechanism of cholera epidemics.** Note: The symbols "+" (positive) and "-" (negative) in the figure indicate the expected direction of influence for each factor.

**Table 1. Definitions and Processing Instructions for All Variables in the Model.**

| Variable Name | Variable Definition and Description | Unit | Time Coverage and Processing Method | Role in the Model |
|---|---|---|---|---|
| Cholera Epidemics | The cumulative total of confirmed cholera cases reported by each county between 1949 and 2020 reflects the historical cumulative intensity of cholera epidemics in the region and serves as a core observational indicator for research. | case | Complete study period from 1949 to 2020, utilizing cumulative confirmed case counts from county health statistics reports and disease surveillance systems, with duplicate reports excluded. | Dependent variable, representing the outcome variable of a cholera epidemic |
| Population Density | The density of regional population distribution reflects the concentration of susceptible individuals and serves as a key demographic indicator influencing the transmission efficiency of infectious diseases. | Persons/km² | Based on the data from 7 census years (1953, 1964, 1982, 1990, 2000, 2010, and 2020), the county-scale data were supplemented by the spatial interpolation method, and the average value during the study period was calculated. | Endogenous variable, serving as a proxy variable for the size of susceptible population |
| Annual mean summer temperature | The average temperature from June to August (summer) each year reflects the cumulative summer heat level of a region and affects the survival and reproduction efficiency of Vibrio cholerae. | °C | For the complete study period from 1949 to 2020, county-scale data were extracted through spatial overlay analysis based on monthly temperature raster data, and the annual average values during the study period were calculated. | Endogenous variable, serving as a proxy variable for the living environment of Vibrio cholerae (infectious source) |
| River Network Density | The ratio of the total length of rivers to the regional area within a county reflects the degree of water system development and the spatial characteristics of water body coverage, and serves as a key environmental carrier indicator for cholera water-borne transmission. | km/km² | As static geographic data, river elements within county boundaries were extracted based on the 1:100,000 water system vector data from Open Street Map (OSM), and the total river length per unit area was calculated. | Endogenous variable, serving as a proxy variable for the water-borne transmission route of cholera |
| Annual Mean GDP | The average annual gross domestic product (GDP) within a county, which reflects the regional economic development level, the capacity for public health resource allocation, and the potential for medical prevention and control investment. | ten thousand CNY | Actual measurement data from 1949 to 2020 (sourced from statistical yearbooks), with annual averages calculated for the study period. | Exogenous variable, serving as a core indicator representing the level of social and economic development |
| Night Lights | The annual average nighttime light radiation intensity within a county reflects the regional urbanization level, intensity of human activities, and the level of infrastructure improvement | nW/(cm²·sr) | From 1985 to 2020, based on the DMSP-OLS (Defense Meteorological Satellite Program-Operational Linescan System) nighttime light dataset, the county-level average values were calculated after conducting data consistency correction. | Exogenous variable, serving as a proxy variable for urbanization and human activity intensity |
| Road Density | The ratio of the total length of highways/roads within a county to the regional area reflects traffic accessibility, the convenience of personnel mobility, and the efficiency of public health emergency response. | km/km² | Covering 6 time nodes (1962, 1980, 1990, 2000, 2010, and 2020), county-level highway networks were extracted based on transportation vector map data, and the average value of total highway length per unit area was calculated | Exogenous variable, an indicator representing traffic connectivity and emergency response capacity |
| Mean Elevation | The average value of topographic elevation within a county reflects the characteristics of regional terrain relief, affects climatic conditions and human settlement patterns, and indirectly acts on cholera transmission | m | As static geographic data, it was obtained through county boundary clipping and mean value statistics based on the 30m-resolution SRTM DEM | Exogenous variable, a basic indicator representing natural geographical characteristics |
| Annual Mean Precipitation | The annual average precipitation within a county reflects the characteristics of regional aridity and humidity, affects the stability of the aquatic environment and the transmission vector conditions of Vibrio cholerae. | mm | For the complete study period from 1949 to 2020, county-scale data were extracted via spatial overlay analysis based on the 0.5° × 0.5° monthly precipitation raster data, and the annual average values over the study period were calculated. | Exogenous variable, a key indicator representing natural climatic characteristics |

*(Continued)*

**Table 1.** (Continued)

| Variable Name | Variable Definition and Description | Unit | Time Coverage and Processing Method | Role in the Model |
|---|---|---|---|---|
| Annual Mean Sunshine | The annual average value of sunshine duration in a county is able to reflect the degree of abundance of solar radiation resources in the region | hour | From 1960 to 2020, based on the monthly sunshine raster data generated from meteorological station observation data and spatial interpolation, the annual average values at the county scale over the study period were calculated | Exogenous variable, an indicator representing climatic conditions |
| Years of Flooding | The cumulative frequency of flood disasters within counties from 1949 to 2000 reflects the intensity of water inundation events caused by heavy rainfall, snowmelt and other factors, and is prone to triggering waterborne transmission of cholera | frequency | From 1949 to 2000, flood disaster events with clear time records within counties were counted based on historical disaster records | Exogenous variable, an indicator representing disaster disturbance factors |
| Years of Drought | The cumulative frequency of drought disasters in counties from 1949 to 2000 reflects the intensity of water shortage events caused by long-term insufficient precipitation, and affects sanitary conditions and disease prevention and control | frequency | From 1949 to 2000, drought disaster events with clear time records within counties were counted based on historical disaster records | Exogenous variable, an indicator representing disaster disturbance factors |

Note: This model adopts a time aggregation strategy, where all variables are processed into a representative value for each county-level unit over the study period (1949–2020), forming a cross-sectional data structure for structural equation model (SEM) analysis.

(3) Data Quality and Limitations. Although the cholera epidemics data in this study have undergone multi-source verification, they still have certain limitations due to the constraints of technical conditions and objective environments in specific historical periods. The core influencing factors of data quality stem from the developmental differences in cholera surveillance systems across different eras, with specific manifestations as follows: Before the 1970s, the surveillance system was initially established and relied on manual case reporting. The data underreporting rate was high, and only the rough scope of the epidemics could be grasped, resulting in prominent limitations. During the 1980s-1990s, the surveillance network was gradually taking shape, and the sentinel system for intestinal clinics was fully popularized, which significantly improved the standardization and completeness of surveillance data. However, problems such as insufficient accuracy of strain typing still existed. Since the 21st century, the introduction of molecular biology technologies such as PCR and whole-genome sequencing has achieved a leap-forward improvement in the accuracy of surveillance data, providing core support for strain tracing and transmission chain analysis. With the real-time uploading and integration of data, the underreporting rate at the national level has dropped to below 1%. Nevertheless, insufficient grassroots testing capacity in remote areas remains a shortcoming. In general, China's cholera surveillance system has shown a staged evolution characteristic of "from passive response to active intelligent early warning", and the data quality has undergone significant changes with socio-economic development and technological innovation [42].

The selection of influencing factors in this study was guided primarily by the theoretical framework, while also taking into account the principle of data availability. Constrained by the consistency and continuity of historical data series at the county scale, there were significant discrepancies in the completeness of datasets for various influencing factors: Among them, temperature, precipitation, and economic data could be continuously obtained at the county scale throughout the entire study period, with high data completeness; In contrast, only cross-sectional data at breakpoint of each era were available for population density and road network density datasets, making it impossible to form continuous time series. Data on drought and flood disaster sequences were missing after 2000; sunshine duration data were absent for the period

1949–1959; and nighttime light data were not available from 1949 to 1984; given the strong stability of topography and river-lake distribution, elevation and river network data from 2020 were used as substitutes. In addition, some factors that have been proven to affect cholera epidemics could not be quantitatively incorporated into the model due to data limitations. For example, vaccination coverage, water supply and sanitation facility conditions, etc., could not be directly included in statistical analyses due to the lack of standardized quantitative data at the county scale.

## Methods

**Mann-Kendall Test (M-K Test).** The Mann-Kendall Test (M-K Test) is a nonparametric method widely used in climate, hydrology, and other fields to analyze trends and detect abrupt changes in time series data. Due to its robustness to non-specific data distributions and insensitivity to missing values [43], the M-K test is widely applied in trend analysis and mutation detection for long-term infectious disease sequences [27,44,45]. Spanning 72 years, this study's early data might exhibit discontinuities and recording biases; the M-K test reliably identified overall trends and statistical significance in cholera incidence rates. The calculation method and formula are:

For a time series $x_i (i = 1,2,...,n)$ with $n$ sample sizes, construct an order column $S_k$, $S_k$ denotes the cumulative number of $x_i > x_j$ ($1 \leq j \leq i$) in the ith sample, and define $S_k$ as:

$$S_k = \sum_{i=1}^{k} r_i (k = 1, 2,..., n), r_i = \begin{cases} 1 & x_i > x_j \\ 0 & x_i \leq x_j \end{cases} (j = 1, 2,..., i)$$

(1)

Assuming that the time series are independently randomized and the mean and variance of $S_k$ are $E(S_k)=k(k-1)/4$,- $Var(S_k)= k(k-1)(2k+5)/72$, respectively, the formula for the statistic $UF_k$ is defined as:

$$UF_k = \frac{S_k - E(S_k)}{\sqrt{Var(S_k)}} (k = 1, 2, 3,...n)$$

(2)

Where $UF_k$ and $UB$ are sequences of statistics calculated based on the time series $x$ in the order $x_1, x_2, ..., x_n$ and the inverse order $x_n, x_{n-1}, ..., x_1$, both of which obey the standard normal distribution. If the value of $UF$ is greater than 0, it indicates an upward trend in the sequence; less than 0 indicated a downward trend. When the value of $UF$ or $UB$ exceeds the critical confidence level straight line (the confidence level line is ±1.96 at the test confidence level α = 0.05), it indicates that the upward or downward trend was significant, and the segment above the critical line is identified as the time region in which the mutation occurred. In this study, Matlab software was used to program and plot the statistics UB and UF of the time series of the number of cholera cases in Hubei Province to test the trend of the cholera epidemics and the time point of mutation in Hubei Province.

**Wavelet Analysis.** Wavelet Analysis is a new branch of mathematics formed in the late 1980s, which is widely used in many fields such as signal processing because of its good time-frequency localization analysis capability [46]. The basic idea of wavelet analysis is to use a cluster of wavelet function systems to represent or approximate a signal or function. The wavelet function is the key to wavelet analysis, and there are many wavelet functions that could be selected. This paper selectes the Morlet wavelet function, primarily based on two considerations. First, the Morlet wavelet exhibits good localization properties in both the time and frequency domains, allowing for an optimal balance between temporal reso-lution and frequency resolution. This makes it particularly suitable for analyzing non-stationary time series such as chol-era case counts [27,44,45]. Second, its oscillatory waveform better matches potential quasi-periodic fluctuation signals within the series, enabling effective extraction of latent periodic components. Compared to the Daubechies wavelet, which is more adept at capturing abrupt changes, or the Mexican hat wavelet (Ricker wavelet), which has relatively weaker frequency-domain resolution, the Morlet wavelet is better aligned with the analytical objectives of this study. In terms of

parameter settings, continuous wavelet transform was employed, with the scale range corresponding to periods from 2 to 35 years, to detect potential fluctuations from short-term to long-term cycles. The scales were densely sampled using a logarithmic equal-interval approach across 64 scale levels to ensure the continuity of the periodogram, thereby enabling precise identification of the dominant cycles in the series. By computing the real part of the wavelet coefficients, the oscillation phase and intensity of the series across different time scales can be analyzed; meanwhile, calculating the wavelet variance allows for the determination of the dominant periodic scales. In this paper, Matlab software's wavelet analysis tool was used to calculate the real part of wavelet coefficients and wavelet variance of the number of cholera cases in Hubei Province, and Surfer software was used to draw the contour map of the real part of wavelet coefficients.

**Geographic Information System (GIS) Spatial Analysis.** (1) Spatial Autocorrelation Analysis. Spatial Autocorrelation Analysis can measure the spatial distribution pattern of an outbreak based on the similarity of spatial locations and attributes. It can also detect spatial clustering and anomalies, including both global and local spatial autocorrelation. Global spatial autocorrelation determines whether the spatial distribution of an outbreak has an overall spatial correlation and the magnitude of the correlation by using Moran's $I$. Moran's $I$ is calculated by the formula:

$$Moran's I = \frac{n \sum_{i=1}^{n} \sum_{j \neq 1}^{n} W_{ij}(x_i - \overline{x})(x_j - \overline{x})}{\sum_{i=1}^{n} \sum_{j \neq 1}^{n} W_{ij} \sum_{i=1}^{n} (x_i - \overline{x})^2} \tag{3}$$

Local Spatial Autocorrelation Analysis can further clarify the degree of spatial dependence and clustering pattern of the epidemic on the local space. The $I_{LISA}$ is used for statistical purposes, and hypothesis testing is performed to obtain the value of $Z$. The $I_{LISA}$ is calculated by the formula:

$$I_{LISA} = \frac{n(x_i - \overline{x})}{\sum_{i=1}^{n} (x_i - \overline{x})^2} \sum_{j=1}^{n} w_{ij}(x_j - \overline{x}) \tag{4}$$

Where $n$ represented the number of spatial units, i.e., 103 counties in Hubei Province; $x_i$ and $x_j$ referred to the number of cholera cases in counties $i$ and $j$, respectively; $\overline{X}$ represented the mean number of cholera cases in all counties; and $W_{ij}$ was the spatial weight of counties $i$ and $j$. In this study, the ArcGIS "spatial autocorrelation" tool was used to calculate Moran's $I$ for global spatial autocorrelation analysis, and the "clustering and outlier analysis" tool was used to analyze local spatial clustering patterns, and when generating the spatial weight matrix, the inverse distance was adopted to define spatial relationships.

(2) Distributional Center of Gravity Change Analysis. Referring to the population center of gravity model [47], the center of gravity of epidemic distribution is the point in the spatial plane where moments reach equilibrium, based on the distribution of cases in a given region. The equilibrium of the epidemic distribution can be assessed by comparing this center of gravity with the geometric center of the region. Analyzing the center of gravity of cholera epidemics in Hubei Province at different times can clarify the spatial distribution and development trend of the epidemics, which is calculated as:

$$\overline{X} = \sum_{i=1}^{n} w_i x_i / \sum_{i=1}^{n} w_i, \overline{Y} = \sum_{i=1}^{n} w_i y_i / \sum_{i=1}^{n} w_i \tag{5}$$

Where $\overline{X}$ and $\overline{Y}$ center of gravity coordinates that made up the cholera epidemics in the study area ($\overline{X}$, $\overline{Y}$), $n$ represented the total number of study units; $x_i$ and $y_i$ constituted the center of gravity coordinates of the $i$th study unit($x_i$, $y_i$), $w_i$ was the number of cholera cases in the ith study unit. In this study, the ArcGIS "Mean Center" tool was used to calculate the center of gravity of cholera epidemics in Hubei Province at different time periods using the number of years of cholera as weights.

**Structural Equation Model (SEM).** Structural Equation Model (SEM) is a statistical method based on the covariance matrix of variables that analyzes the relationship between variables (including the role of exogenous variables on endogenous variables and the relationship between endogenous variables, etc.). It integrates multivariate statistical analyses, such as path analysis, factor analysis and regression analysis [48]. In this study, only the path relationships between exogenous and endogenous variables were considered, and latent variables were not included, so the structural equation modeling could be expressed as follows:

$$Y=BY+\Gamma X + \zeta \tag{6}$$

where $Y$ is an endogenous variable, represented by a vector of endogenous indicators, and $X$ is an exogenous variable, represented by a vector of exogenous indicators; B is the matrix of regression coefficients between endogenous variables; $\Gamma$ is the matrix of regression coefficients between exogenous and endogenous variables; $\zeta$ is the regression residual, which reflects the part of $Y$ that is not explained in the equation. The ratio of chi-square degrees of freedom (CMIN/DF), root mean square of approximation error (RMSEA), and goodness-of-fit index (GFI) are used to evaluate the goodness-of-fit of the model. In this study, structural equation modeling was conducted using the Amos plug-in in SPSS 27.0 to analyze the causal relationships among environmental factors and the mechanisms of action on cholera epidemics through path diagrams and effect values. Among these, the effect value can be further decomposed into direct and indirect effects. Table 4 presents precisely this decomposition, interpreted as follows: The direct effect refers to the strength of a variable's direct influence on another variable. It manifests as the standardized path coefficient between the two variables after controlling for all other variables in the model, reflecting the "net effect" of the independent variable on the dependent variable. For example, after controlling for variables such as the average annual summer temperature and river network density, the path coefficient from "population density" to "level of cholera epidemics" represents the direct effect of population density on cholera outbreaks. Indirect effects refer to the cumulative impact of a variable on the dependent variable through one or more mediating variables, calculated as the sum of the products of the standardized coefficients across all mediating paths. For instance, "road network density" may influence cholera epidemic levels by first increasing "population density" (path: road network density→population density), which then affects cholera epidemic levels (path: population density→cholera epidemic levels). The combined effect generated by this transmission path constitutes the indirect effect of road network density. The total effect is the sum of the direct effect and all potential indirect effects, representing the overall strength of the independent variable's influence on the dependent variable through all defined pathways in the model. This satisfies the quantitative relationship: Total Effect = Direct Effect + Indirect Effect. This study employs a time-averaging approach to construct a cross-sectional dataset, aiming to identify the steady-state drivers shaping the long-term spatial patterns of cholera rather than short-term volatility triggers. By focusing on spatial variation, it aligns with the application premise of structural equation modeling in identifying long-term drivers.

## Results

### Temporal changes in the cholera epidemics in Hubei Province

See Fig 3.

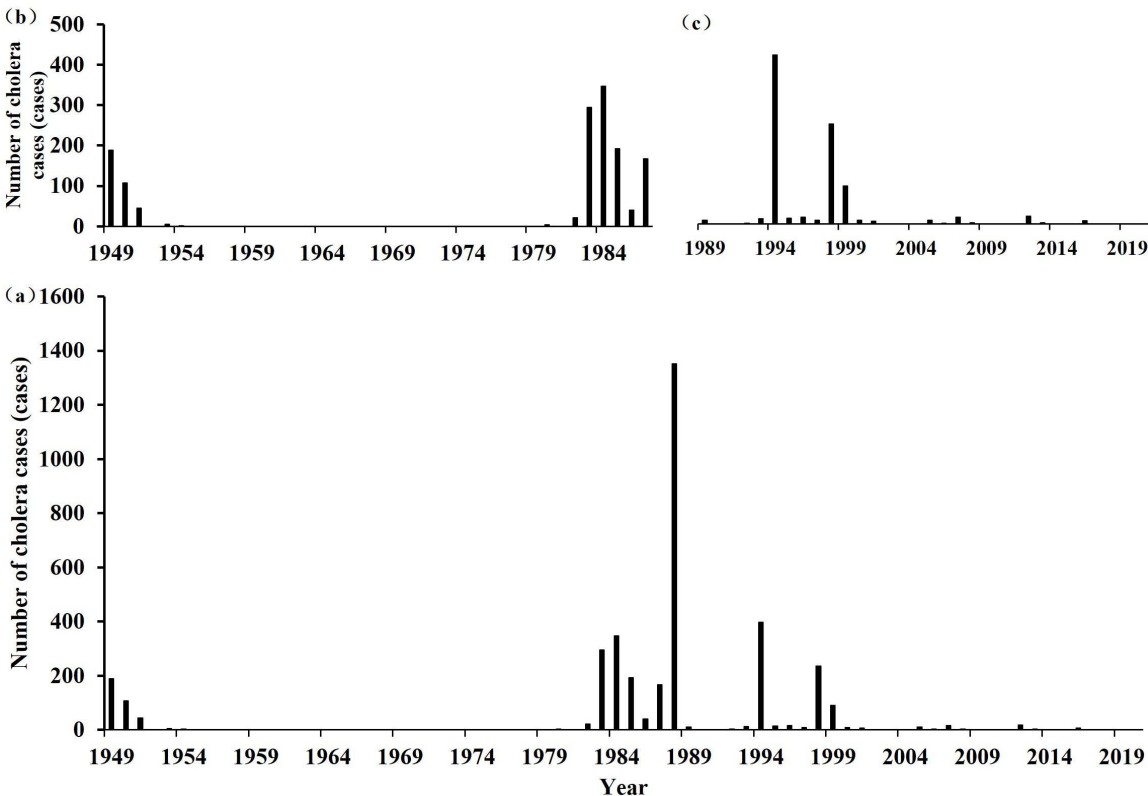

**Fig 3. Changes in the number of cholera cases in Hubei Province from 1949 to 2020.** (a)1949—2020; (b)1949—1987; (c)1989—2020.

## Trends

From 1949 to 2020, over a span of 72 years, cholera was reported in Hubei Province in 31 of those years (Fig 3), with a cumulative total of 3,608 cases. The years with the highest number of cholera cases were 1988 (1,351 cases), 1994 (396 cases), 1984 (347 cases), 1983 (294 cases), and 1998 (204 cases). To facilitate the description of the macroevolution of cholera in Hubei Province from a historical epidemiological perspective, we first divided the period from 1949 to 2020 into four descriptive phases based on the distribution characteristics of case numbers, important social and historical nodes, and transitions in disease prevention and control stages. 1949–1954 (Low Incidence Period): In the early years after the founding of the People's Republic of China, cholera cases in Hubei Province were sporadic, with a cumulative total of 345 reported cases, marking a relatively low-incidence phase. 1955–1979 (Cholera-Free Period): Hubei Province experienced no cholera cases for 25 consecutive years, representing a disease-free phase. 1980–1999 (High Incidence Period): This marked the first major cholera epidemic after 1949, with a cumulative total of 3,193 cases, accounting for 88.4% of all cholera cases in Hubei Province, making it a high-incidence phase. 2000–2020 (Controlled Incidence Period): Cholera epidemics were gradually brought under control, with the cumulative number of cases dropping to just 70, indicating a controlled-incidence phase.

A Mann-Kendall (M-K) test was performed on the time series of cholera cases in Hubei Province from 1949 to 2020 (Fig 4). The UF curve shows that after 1955 (the start of the cholera-free period), the statistic remained in the negative range for a long time and fell below the significance threshold of -1.96 multiple times between 1955 and 1994. Among these, the 25-year consecutive cholera-free period (1955–1979) at the beginning of the series profoundly influenced the

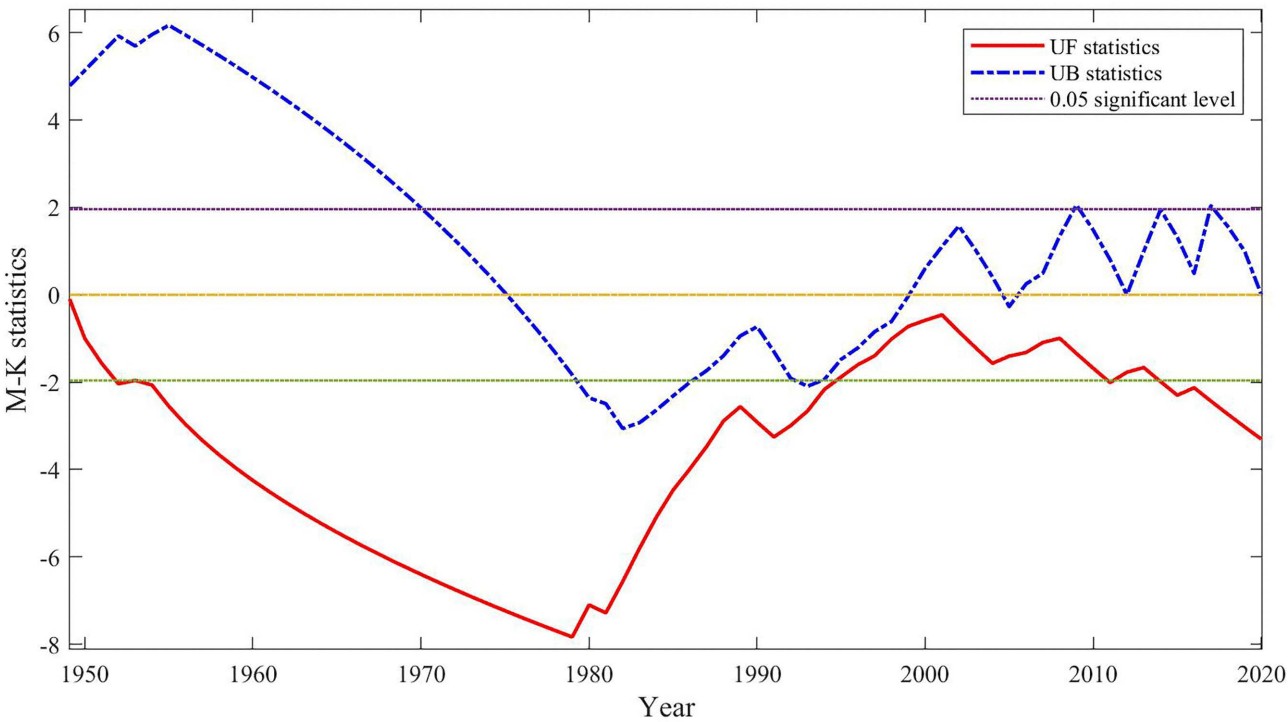

**Fig 4. Mann-Kendall test of cholera cases in Hubei Province from 1949 to 2020.**

trend judgment. A negative UF value indicates that, from the perspective of the long-term baseline state, after the number of cases rose sharply in the 1980s, its overall level did not continue to move toward higher extreme values but instead showed a statistical tendency to decline from the peak range. Specifically, the rate of decline was relatively fast during 1955–1978 (corresponding to the cholera-free period) and gradually slowed during 1979–1994 (which included the epidemic outbreak and high-level fluctuation period). After 1995, the UF curve fluctuated within the non-significant range until it exhibited a significant downward trend again from 2013 onward, which is consistent with the actual situation that the epidemic entered a "low-incidence and controllable phase" after 2000. No intersecting abrupt change points were observed between the UF and UB curves, indicating that the long-term overall trend of cholera incidence showed a downward tendency throughout the entire study period from 1949 to 2020.

## Cyclical variations

The real part of the Morlet wavelet transform coefficients reflects the fluctuation characteristics of data within the wavelet domain across different time scales. The alternating positive and negative values of the real part correspond to the severity characteristics of the cholera sequence in the time domain. Plotting the isocontour map of the real part of the wavelet transform coefficients for the cholera epidemic sequence (Fig 5), the colors represent the sign and magnitude of the real part of the wavelet coefficients: red regions indicate positive real parts of the wavelet coefficients, corresponding to periods of elevated cholera case numbers; blue regions indicate negative real parts of the wavelet coefficients, corresponding to periods of reduced case numbers. It can be observed that the cholera epidemics in Hubei Province exhibited different periodic variations and alternating high-low incidence processes across different time scales, forming oscillation centers with alternating positive and negative values at various scales. Overall, the cholera epidemics in Hubei Province from 1949 to 2020 showed fluctuation cycles at three time scales: 25-35a, 16-22a, and 5-10a. Specifically: At the 25-35a time scale,

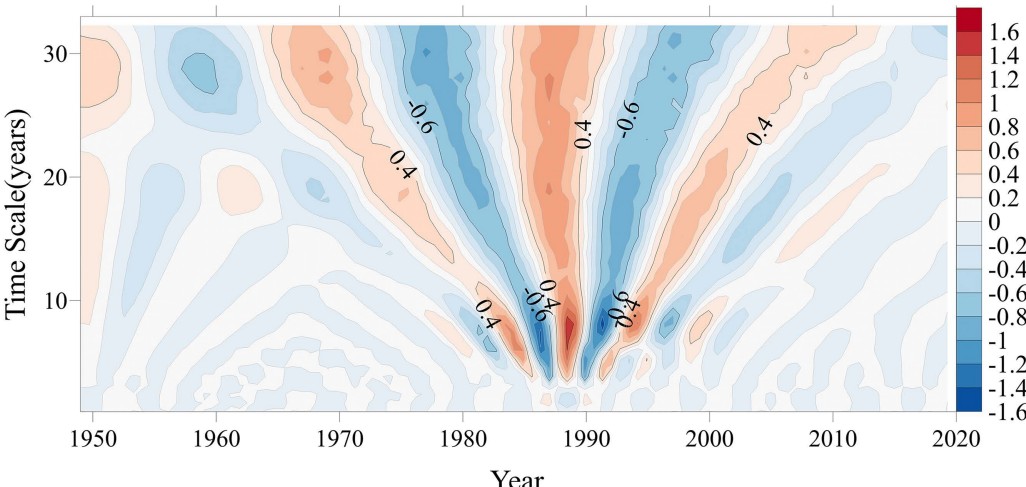

**Fig 5. Real contour plot of wavelet transform coefficients of cholera cases in Hubei Province.**

there were 4 quasi-oscillation cycles with alternating high and low incidence, which were relatively stable and exhibited full-domain coverage throughout the entire study period. At the 16-22a time scale, there were 4 quasi-oscillation cycles with alternating high and low incidence, which mainly occurred between 1960 and 2005. At the 5-10a time scale, there were 4 quasi-oscillation cycles with alternating high and low incidence, which mainly occurred between 1980 and 1999.

Wavelet variance maps can reflect the distribution of fluctuation energy of cholera sequences over time scales, and can be used to determine the existence of multilevel main cycles of cholera sequences. Plotting the wavelet variance map of cholera sequences in Hubei Province (Fig 6), it could be seen that there were three obvious peaks in the cholera

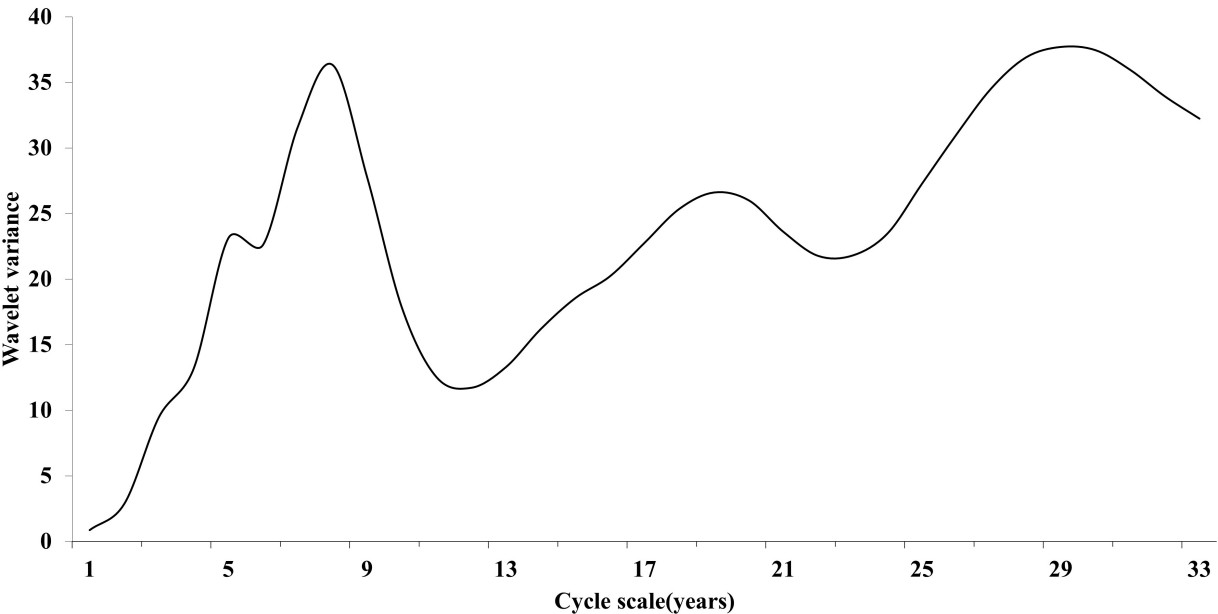

**Fig 6. Wavelet Variance Plot of Cholera Case Count Time Series in Hubei Province.**

sequences in Hubei Province between 1949 and 2020, which corresponded to the time scales of 29, 19, and 8 years in turn. Among them, the largest peak corresponds to the 29a time scale, indicating that the cyclic oscillation around 29 years was the strongest and serves as the first main cycle of cholera variations in Hubei Province; the 8a and 19a time scales were the second and third main cycles, respectively. These three scales of cycle fluctuations controled the characteristics of cholera changes throughout the time domain.

According to the results of the wavelet variance test, plotting the process lines of the real part of the wavelet transform coefficients of the 29a time scale and 8a time scale main cycles of the cholera case count series in Hubei Province during 1949–2020 (Fig 7), we can analyze the characteristics of the mean cycle and light-heavy change that existed during the process of the cholera change in Hubei Province. On the 29a time scale, the mean change cycle of the number of cholera cases was about 17a, and it had experienced about 4 cycles of light-heavy transition, with the periods of heavy epidemics: 1949–1954, 1964–1972, 1983–1992, 2003–2012, and the periods of light epidemics: 1955–1963, 1973–1982, 1993–2002, and 2013–2020. On the 8a time scale, the mean change cycle of the number of cholera cases is about 7a, and it has experienced about 10 distinct cycles of light-heavy transition, with the periods of heavy epidemics: 1949–1950, 1954–1956, 1972–1974, 1978–1979, 1983–1984, 1988–1989, 1993–1995, 1998–2000, 2004–2005, 2009–2013, and the periods of light epidemics: 1951–1953, 1957–1959, 1975–1977, 1980–1982, 1985–1987, 1990–1992, 1996–1997, 2001–2003, 2006–2008, 2014–2016.

## Seasonal changes

Seasonal distribution characteristics of cholera epidemics were analyzed based on cholera data with explicit seasonal information. In the 70 years from 1949-2020, there were cholera epidemics in spring in a total of 2 years (5.13%), cholera occurred in summer in a total of 17 years (43.58%), cholera occurred in autumn in a total of 19 years (48.72%), and

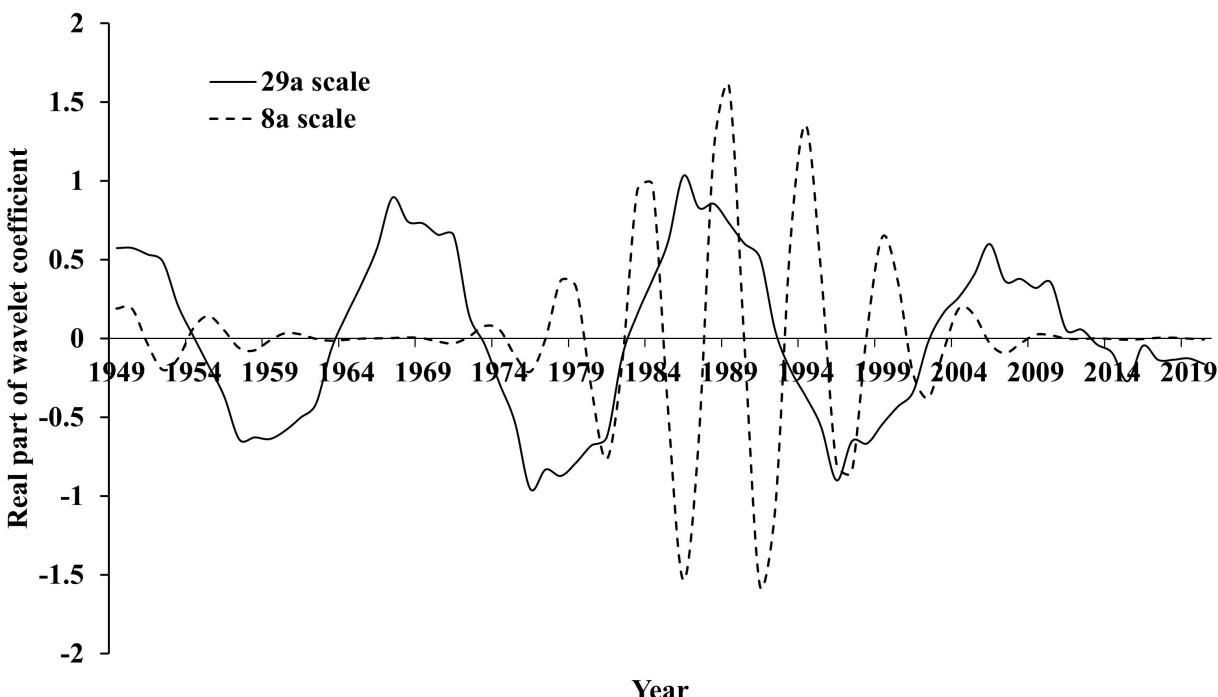

**Fig 7. Process lines of the real part of wavelet transform coefficients of the number of cholera cases in Hubei Province under 29a time scale and 8a time scale cycles.**

cholera occurred in winter in a total of 1 year (2.56%). It could be seen that since 1949, summer and autumn were the high incidence seasons of cholera, while spring and winter were more rare. Counting the number of cases of cholera epidemics in each season, excluding those with unknown seasons, there were a total of 3 cases (0.19%) in spring, 759 cases (49.19%) in summer, 773 cases (50.1%) in autumn, and 8 cases (0.52%) in winter. It showed that summer and autumn were the seasons with the greatest impact of cholera, precisely by month, with the highest number of cases in August, September, and October, with 487 cases (31.56%) in August, 510 cases (33.05%) in September, and 236 cases (15.29%) in October. Combining the seasonal frequency of cholera epidemics and the number of incidence statistics, cholera epidemics in Hubei Province from 1949 to 2020 mainly occurred in the summer and fall seasons, especially in the months of August, September and October, and were rare in spring and winter.

### Spatial distribution of cholera epidemics in Hubei Province

**Overall distribution characteristics.** Over the 72-year period from 1949 to 2020, among the 103 county-level administrative units in Hubei Province, a total of 71 counties/districts reported cholera epidemics, accounting for approximately 70% of the total. The cumulative number of cholera cases reached 3,608. The hardest-hit areas were mainly concentrated in Daye (344 cases) and Xian'an (326 cases) in the Xianning region, followed by Jianli (168 cases), Jiangling (143 cases), and Hanchuan (141 cases). Light or non-epidemics areas were primarily located in western Hubei Province, including Shiyan, Shennongjia, and Enshi (Fig 8a). Spatial autocorrelation analysis revealed that the cholera epidemics in Hubei Province exhibited a significant clustered distribution pattern, with a Moran's *I* value of 0.35, a Z-score of 6.35 (p < 0.01). Local Indicators of Spatial Association (LISA) analysis (Fig 8b) further accurately identified the spatial locations and types of local hotspots and coldspots. High-high (HH) agglomeration areas were mainly continuously distributed in the core urban districts of Wuhan, such as Wuchang, Hanyang, and Jiangxia, as well as in Xianning City in the southeast (including Xian'an, Chongyang, and Yangxin). In contrast, low-low (LL) agglomeration areas were concentrated in Shiyan City, Shennongjia Forestry District, and most parts of Enshi Tujia and Miao Autonomous Prefecture in western Hubei Province.

**Spatial distribution changes.** In the early years of the People's Republic of China (1949–1959), cholera epidemics were relatively scattered (Fig 9a), primarily concentrated in eastern Hubei Province. Among the 103 counties, 30 reported cholera cases, with Xiaonan being the most severely affected (60 cases), followed by Hanchuan (54 cases), Anlu (30

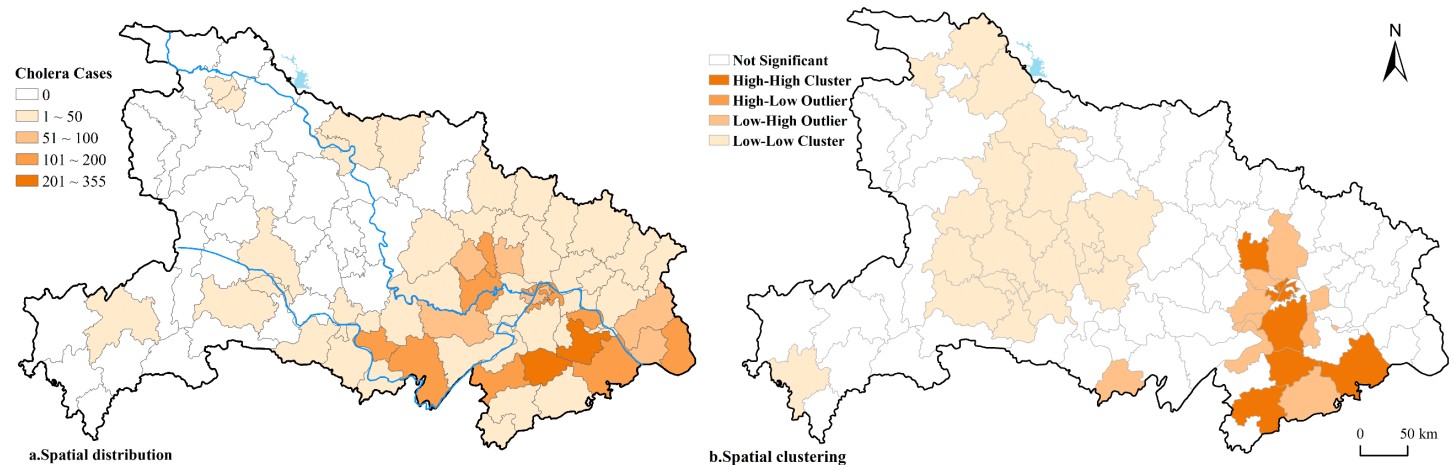

**Fig 8. The overall distribution characteristics of cholera epidemics in Hubei Province from 1949 to 2020.** Note: The basemap came from United States Geological Survey (https://apps.nationalmap.gov/services/), the map boundary has not been changed. Cartographic software: ArcGIS.

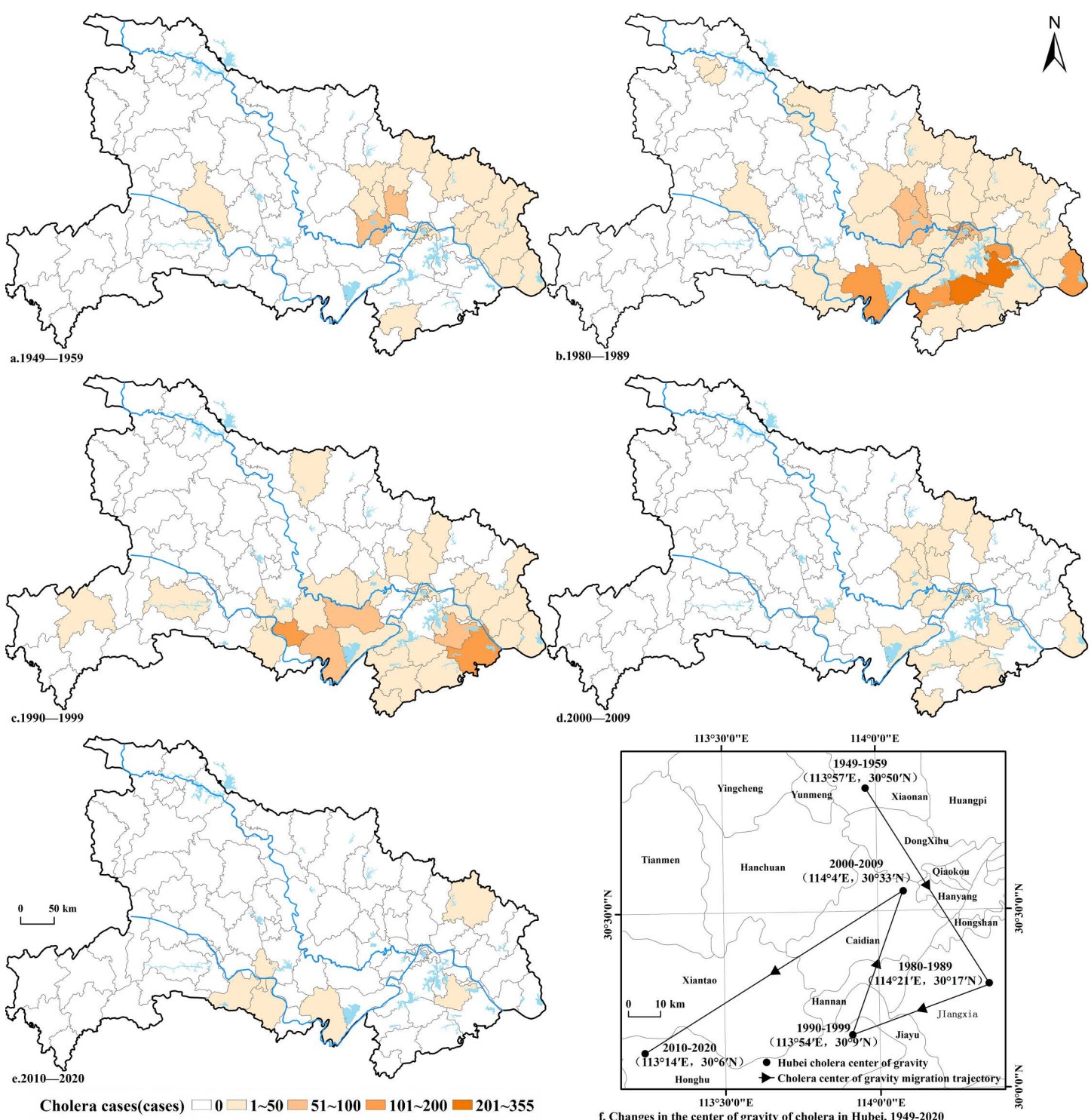

**Fig 9. Cholera epidemics in Hubei Province by chronology from 1949 to 2020.** Note: The basemap came from United States Geological Survey (https://apps.nationalmap.gov/services/), the map boundary has not been changed. Cartographic software: ArcGIS..

cases), and Dawu (30 cases). Subsequently, no cholera cases were reported in Hubei Province for the next two decades until 1980, when three imported cases emerged. From 1980 to 1989, cholera epidemics intensified sharply (Fig 9b), spreading to 57 counties—about 50% of the province. The most severe outbreaks occurred in Wuhan's urban districts (Qiaokou, Hanyang, Hanjiang, Hongshan, Jiang'an, Qingshan, Wuchang), where cholera was reported in eight out of ten years, with a total of 627 cases. Xianning and Daye followed, with 314 and 279 cases, respectively. Nearly all counties in eastern Hubei Province were affected. From 1990 to 1999, cholera remained a serious public health issue (Fig 9c), with outbreaks in 42 counties and a total of 739 cases. Jiangling County was the hardest-hit area, and infections were mainly distributed along the Yangtze River and the lower reaches of the Han River. From 2000 to 2009, cholera cases declined (Fig 9d), with only 23 affected counties and 41 reported cases, primarily concentrated in Wuhan's urban areas, its surroundings, and southeastern Hubei Province. From 2010 to 2020, cholera epidemics in Hubei Province were effectively controlled, with only sporadic cases reported in five counties: Macheng, Gong'an, Songzi, Jingzhou, and Huangshi.

The shifting centroid of cholera epidemics can reflect the spatial distribution and evolving trends of the epidemics (Fig 9f). From 1949 to 2020, the cholera centroid in Hubei Province oscillated along the eastern edge of the Jianghan Plain. Initially located within Xiaonan, it later shifted southward to Jiangxia, Honghu, and Caidian before moving southwestward back to Honghu, exhibiting a general migration trend from northeast to southwest.

### The formation mechanism of the cholera epidemics in Hubei Province

**Construction and validation of a cholera epidemics impact mechanism model.** Prior to constructing the structural equation model, a multicollinearity diagnosis was conducted for all variables in the model to ensure the stability and validity of parameter estimation. The variance inflation factor (VIF) and tolerance were calculated using SPSS's linear regression procedure as judgment criteria. The test results (Table 2) showed that the VIF values for all independent variables were below 10, and the tolerance values were all greater than 0.1. Therefore, it could be concluded that no severe multicollinearity issues exist in the model, and the regression results are stable and reliable. Based on this, natural factors such as annual mean precipitation, mean elevation, and annual mean sunshine at the county level, human factors such as annual mean GDP, night lights, and road network density, as well as disaster factors including floods and droughts, were selected as exogenous variables. Population density, annual mean summer temperature, and river network density were chosen as endogenous variables. The structural equation model was analyzed using Amos software, and the model was fitted using the maximum likelihood method. This process elucidated the mechanisms and

**Table 2. Results of Variance Inflation Factor (VIF) and Tolerance Tests.**

| Factor | VIF | Tolerance |
| --- | --- | --- |
| Annual mean summer temperature | 1.131 | 0.884 |
| Annual mean precipitation | 1.917 | 0.522 |
| Mean elevation | 2.697 | 0.371 |
| Annual mean sunshine | 2.309 | 0.433 |
| River network density | 2 | 0.5 |
| Road density | 1.066 | 0.936 |
| Population density | 5.643 | 0.177 |
| GDP | 2.253 | 0.444 |
| Night Lights | 7.527 | 0.133 |
| Flood disaster | 2.314 | 0.432 |
| Drought disaster | 2,415 | 0.414 |

pathways through which various environmental factors influence the intensity of cholera outbreaks at the county level in Hubei Province (Fig 10).

From the model fit test results presented in Table 3, it can be observed that the primary fit index, Chi-square (CMIN), is 4.467(p = 0.240), with the Chi-square to degrees of freedom ratio (CMIN/DF) at 1.489, falling within the excellent range of 1–3. The Root Mean Square Error of Approximation (RMSEA) is 0.069, which is within the acceptable limit of 0.08. Additionally, the Goodness of Fit Index (CFI) is 0.999, the Incremental Fit Index (IFI) is 0.999, the Normed Fit Index (NFI) is 0.998, and the Tucker-Lewis Index (TLI) is 0.941, all of which meets or exceeds the excellent benchmark of 0.9. Therefore, a comprehensive analysis of the results indicates that the overall model demonstrates a good fit.

**Formation mechanisms of endogenous variables in cholera epidemics.** Table 4 presents the influence of different geographical environmental factors on the transmission and spread of cholera at the county level. The total effects represent the direct impacts of the endogenous variables (i.e., population density, river network density, and annual

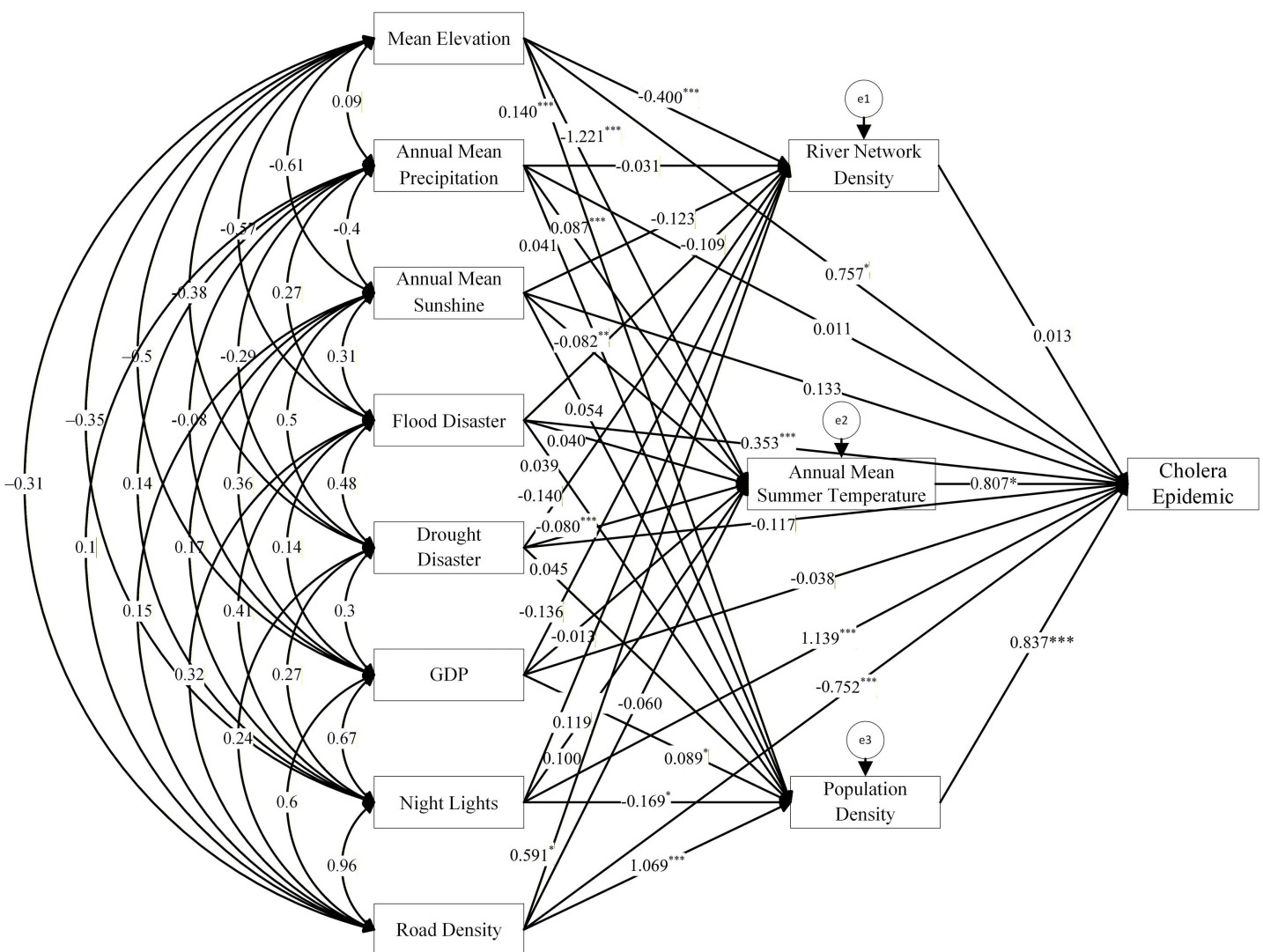

**Fig 10. Mechanism and transmission path of influencing factors of cholera epidemics in Hubei Province.**

**Table 3. Goodness-of-fit-test for the SEM.**

| Indicator | Statistic | Reference Standard | Results |
|---|---|---|---|
| Absolute Compatibility Index | CMIN | P > 0.05, the smaller the better | 4.467($p = 0.240$) |
| | CMIN/DF | 1~3 is excellent, 3–4 is good | 1.489 |
| | RMSEA | <0.05 is excellent, <0.08 is good | 0.069 |
| | CFI | >0.9 is excellent, >0.8 is good | 0.999 |
| Value-added Adaptability Index | IFI | >0.9 is excellent, >0.8 is good | 0.999 |
| | NFI | >0.9 is excellent, >0.8 is good | 0.998 |
| | TLI | >0.9 is excellent, >0.8 is good | 0.941 |

**Table 4. Direct, indirect and overall effects of cholera epidemics in Hubei Province.**

| Factor | Effect | Annual mean summer temperature | River network density | Population density | Cholera epidemics level |
|---|---|---|---|---|---|
| Annual mean summer temperature | overall | – | – | – | **0.807*** |
| River network density | overall | – | – | – | 0.013 |
| Population density | overall | – | – | – | **0.837*** |
| Mean elevation | overall | **-1.221*** | **-0.400*** | 0.140*** | **-0.116*** |
| | direct | **-1.221*** | **-0.400*** | 0.140*** | **0.757*** |
| | indirect | – | – | – | **-0.873*** |
| Annual mean precipitation | overall | **0.087*** | -0.031 | 0.041 | 0.115 |
| | direct | **0.087*** | -0.031 | 0.041 | 0.011 |
| | indirect | – | – | – | 0.104 |
| Annual mean sunshine | overall | **-0.082**** | -0.123 | 0.054 | 0.110 |
| | direct | **-0.082**** | -0.123 | 0.054 | 0.133 |
| | indirect | – | – | – | -0.023 |
| Flood disaster | overall | 0.040 | -0.109 | 0.039 | 0.417*** |
| | direct | 0.040 | -0.109 | 0.039 | 0.353*** |
| | indirect | – | – | – | 0.064*** |
| Drought disaster | overall | **-0.080*** | -0.140 | 0.045 | -0.146 |
| | direct | **-0.080*** | -0.140 | 0.045 | -0.117 |
| | indirect | – | – | – | -0.029 |
| GDP | overall | -0.013 | -0.136 | **0.089*** | 0.024 |
| | direct | -0.013 | -0.136 | **0.089*** | -0.038 |
| | indirect | – | – | – | 0.062 |
| Night Lights | overall | 0.100 | 0.119 | **-0.169*** | **1.080*** |
| | direct | 0.100 | 0.119 | **-0.169*** | **1.139*** |
| | indirect | – | – | – | **-0.059*** |
| Road density | overall | -0.060 | **0.591*** | **1.069*** | **0.102*** |
| | direct | -0.060 | **0.591*** | **1.069*** | **-0.752*** |
| | indirect | – | – | – | **0.854*** |

Note: Values are standardized regression coefficient estimates; *** indicates that the coefficient is significant at the 0.001 level, ** indicates that the coefficient is significant at the 0.01 level, and * indicates that the coefficient is significant at the 0.05 level. All significance tests are two-tailed.

mean summer temperature) on cholera epidemics, with the strength and direction of these effects determined by the estimated standardized regression coefficients. All three endogenous variables exerted positive direct effects on cholera incidence, indicating that increases in population density, river network density, and annual mean summer temperature all contributed to the diffusion and transmission of cholera.

(1) Population density significantly increased the level of cholera epidemics and was the most dominant endogenous factor contributing to cholera epidemics, exhibiting a positive total effect (0.837). Cholera transmission relies on a sufficiently dense human population. Only with population aggregation and urban development could infectious diseases like cholera emerge and propagate [49]. With the increase in population density, the risk of waterborne diseases such as cholera spreading rose when sanitation conditions were poor. Since the founding of the People's Republic of China, particularly after the reform and opening-up, Hubei Province had seen the emergence of several cities with populations exceeding one million, which created potential conditions for cholera epidemics. As shown in Fig 8b, high-value clusters of cholera had been predominantly concentrated in the densely populated towns of Wuhan (Wuchang, Hankou, and Hanyang) and their surrounding regions, while low-value clusters had been mainly distributed in the less populated western Hubei Province region. This pattern demonstrated that cholera epidemics occurred more frequently in areas with higher population density.

(2) The annual mean summer temperature had significantly increased the level of cholera epidemics, with an overall positive effect on cholera epidemics (0.807). Among various climatic and environmental factors, temperature is regarded as one of the key climatic factors determining the survival and reproduction of *Vibrio cholerae* in the natural environment [50]. The optimal growth temperature for *Vibrio cholerae* is 37°C, and the range of 16–42°C is its relatively ideal growth temperature. Exceeding or falling below this temperature range may inhibit its reproduction. The annual mean summer temperature in Hubei Province is approximately 27°C, which is close to the temperature suitable for the reproduction of *Vibrio cholerae*. A rise in temperature is conducive to the rapid reproduction of *Vibrio cholerae*, thereby leading to the prevalence of cholera and triggering cholera epidemics. Cholera in Hubei Province occurred frequently in the three months of August, September, and October when temperatures are relatively high, as well as in the southeastern regions. These months fell in summer and the late summer to early autumn period, during which the annual mean summer temperature was relatively high, which also confirms that the annual mean summer temperature was an important driving factor for cholera epidemics.

(3) As a geographical background factor for epidemic transmission, the path coefficient for river network density was 0.013 (p > 0.05), showing no statistical significance in this study. As a waterborne disease, cholera is primarily transmitted through contaminated well water and river water [12]. Theoretically, densely river-networked areas could provide Vibrio cholerae with more water environments for human contact, thereby creating potential transmission conditions. However, compared to population density and average summer temperatures, the effect size of river network density was lower and not statistically significant. This indicates that within the study model, the river network itself is not the primary statistical driver of epidemics. Instead, it should be regarded as the environmental substrate and necessary precondition for transmission to occur. Nevertheless, the spatial distribution of river network density highly correlates with epidemic clusters (Figs 8 and 9), particularly revealing a spatial coupling pattern of "high river network density—high epidemic clustering" in the densely river-networked Jianghan Plain and its adjacent riverside cities and counties. This indicates that dense river networks provide the geographic foundation for cholera transmission via water sources. Without this condition, outbreaks would struggle to spread through waterways. However, relying solely on dense river networks—without sufficient population density (hosts) and suitable temperatures (promoting pathogen proliferation)—is insufficient to trigger large-scale cholera epidemics.

**Formation mechanisms of exogenous variables in cholera epidemics.** Exogenous variables can not only act directly on the cholera epidemics, but also indirectly by acting on endogenous variables. The overall effect of exogenous variables can be categorized into direct and indirect effects. The direct effect is the estimated value of the standardized regression coefficient of the environmental factor on the level of cholera epidemics, while the indirect effect is the product of the direct effect of the environmental factor on the source of infection, the route of transmission, and the susceptible population, respectively, and the direct effect of the three endogenous variables on the level of cholera epidemics.

(1) Impact of natural environment factors on cholera epidemics.① Elevation had a significant overall negative effect on cholera epidemics (-0.116). In terms of the composition of indirect effects, the average elevation significantly reduces the intensity of cholera epidemics through annual mean summer temperature and river network density, while it significantly enhanced the spread of cholera by affecting population density, with the strongest intensity of action. Hubei Province is located in the transition zone from the second to the third step of China's terrain, with significant topographic differences: the western part is mostly mountainous (high elevation), and the eastern part is mostly plain and hilly (low elevation). This resulted in a dense population in the east and sparse in the west, and cholera epidemics also showed the characteristics of being more cases in the east and less in the west. Topography affected the pattern of epidemics in two ways: first, high elevation reduced population mobility, hindering the spread of cholera; second, increased elevation leads to a drop in temperature, inhibiting the growth and reproduction of *Vibrio cholerae*. Therefore, cholera epidemics were more likely to occur in low-elevation areas.② Precipitation had an overall positive effect on cholera epidemics (0.115), among which the annual average precipitation had the strongest effect on significantly increasing cholera epidemics by influencing the annual mean summer temperature. ③ Sunshine had an overall positive effect on cholera epidemics (0.110), with the positive direct effect greater than the negative indirect effect. In terms of the composition of indirect effects, it significantly reduced cholera epidemics by affecting the annual mean summer temperature. As climatic factors, both precipitation and sunshine had a significant impact on cholera epidemics through temperature. Hubei Province had a subtropical monsoon climate with abundant light energy and sufficient summer precipitation, featuring concurrent rain and heat. High-frequency precipitation in summer triggers large-scale floods, increasing the chance of contact between *Vibrio cholerae* and the population, thereby enhancing the intensity of cholera epidemics. In addition, the overall trend of precipitation in Hubei Province—more in the south and less in the north—had, to a certain extent, established the spatial pattern of epidemics being more severe in southern Hubei Province than in northern Hubei Province. Xianning, the area most severely affected by cholera, was also the area with the most intensive precipitation, while the northwestern Hubei Province, where cholera is less severe, had less precipitation. In general, increased elevation reduced the level of cholera epidemics, while increased precipitation and sunshine hours enhanced the intensity of cholera epidemics.

(2) Impact of disaster factors on cholera epidemics. As the saying goes, "A major epidemic follows a major disaster." [51] Droughts and floods are closely related to the occurrence of epidemics. Disasters alter the living environment of pathogens, providing a hotbed for their growth and reproduction, thereby indirectly inducing the outbreak of plagues [17]. Flood disasters had a significant positive effect on cholera epidemics (0.417), among which the direct positive effect was significant and stronger than the significant positive indirect effect. In terms of the composition of indirect effects, floods had the strongest impact on increasing cholera epidemics by influencing population density. In contrast, drought disasters had a negative effect on cholera epidemics (-0.146), and they significantly reduced the level of cholera epidemics by affecting the annual mean summer temperature. Following floods, damaged or overwhelmed sanitation facilities lead to the overflow of sewage and feces containing Vibrio cholerae, extensively contaminating water sources and creating ideal conditions for the pathogen's proliferation and spread in water

bodies [17]. Simultaneously, densely populated post-disaster shelters with limited sanitation further increase exposure risks and interpersonal contact frequency among susceptible populations. Drought, conversely, limits cholera's waterborne transmission by reducing the volume of water bodies where Vibrio cholerae can survive and spread [11]. In general, flood disasters promoted the incidence and transmission of cholera, while drought disasters inhibited its transmission.

(3) Impact of human factors on cholera epidemics. ① GDP had an overall positive effect on cholera epidemics (0.024), with the negative direct effect being smaller than the positive indirect effect. In terms of the composition of indirect effects, its impact on increasing the epidemics of cholera by influencing population density is the strongest. The economy could indirectly affect cholera epidemics through shaping intermediate variables such as public health conditions, residents' living environments, and medical resources. ② Nighttime light had a significant positive effect on cholera epidemics (1.080), among which the direct positive effect was significant and stronger than the significant negative indirect effect. In the composition of indirect effects, nighttime light had the strongest impact through influencing population density. A core characteristic of nighttime light was its significant correlation with urban population density, intensity of economic activities, and scale of built-up areas. Areas with strong light were often densely populated and industrially agglomerated, with relatively large emissions of domestic and industrial wastewater, thereby indirectly affecting cholera epidemics. ③ Road network density had a significant positive effect on cholera epidemics (0.102), and its impact on cholera through population density was the strongest. The construction of roads and the innovation of transportation tools, on the one hand, strengthened the connections between people and between regions; on the other hand, they also created conditions for the spread of cholera. In general, the significant positive effects of GDP and nighttime light on cholera epidemics meant that a higher level of economic development and road construction played a role in promoting the spread and epidemics of cholera.

**Summary of cholera epidemics formation mechanisms in Hubei Province.**

(1)  The occurrence and spread of cholera epidemics in Hubei Province resulted from the synergistic interaction of natural factors, disaster factors, and human factors, with endogenous variables (population density, annual mean summer temperature, river network density) serving as direct drivers and exogenous variables (e.g., elevation, floods, GDP) exerting indirect effects through regulating endogenous variables. Specifically, population density (the most dominant endogenous factor) and summer temperature jointly amplifed epidemic risk by enhancing susceptible population exposure and *Vibrio cholerae* reproduction, while river network density provided a basic condition for water-borne transmission; exogenous factors further modulate this process—for instance, high elevation inhibits epidemics via dual pathways of lowering temperature and reducing population aggregation, floods exacerbated transmission by polluting water sources, and socioeconomic development (GDP, road density) showed dual effects of promoting population mobility and improving medical prevention capabilities.

(2) The spatiotemporal patterns of Hubei's cholera epidemics were shaped by the multi-scale matching of various influencing factors: ①Temporal dimension: Long-term climate stability (e.g., warm and humid summer conditions) determined the seasonal (summer-autumn) and periodic (55a, 29a main cycles) characteristics of epidemics (Fig 11b, 11c, and 11d); medium-term socio-economic development and pathogen evolution (e.g., introduction of *El Tor Vibrio cholerae* in the 1980s) drived the formation of high incidence period (1980–1999) (Fig 11g); short-term disasters (e.g., 1998 Yangtze River floods) trigger transient epidemic peaks (Fig 11e and 11f). ②Spatial dimension: Macro-scale topography and river networks laid the foundation for the "east-dense, west-sparse" distribution; meso-scale population aggregation and transportation networks (e.g., Wuhan, Xianning) formed regional epidemic hotspots; micro-scale water quality and food hygiene (e.g., post-1980s food pollution replacing water pollution as the main cause) shaped community-level outbreak patterns.

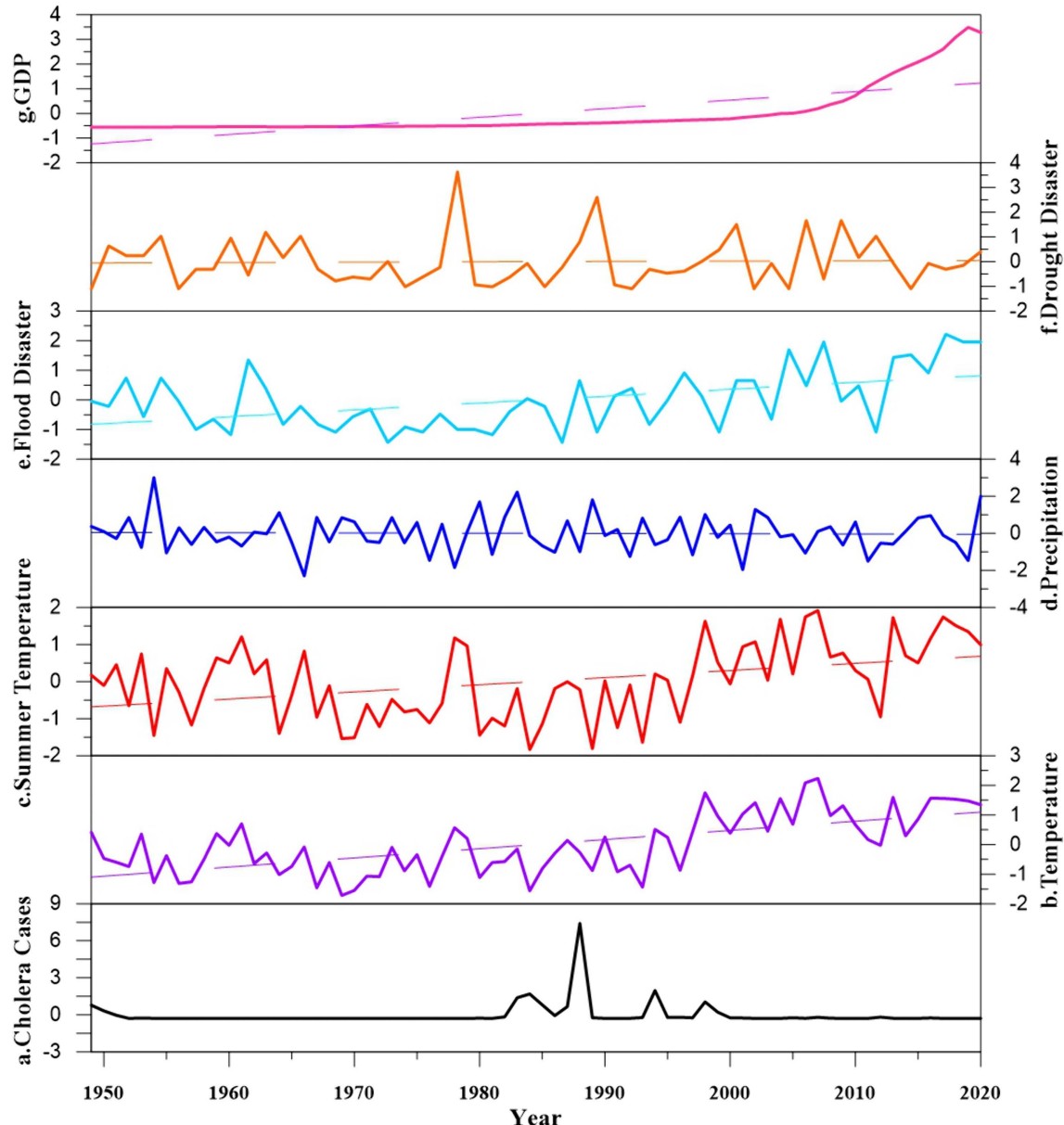

**Fig 11. Comparison of time series between cholera and influencing factors in Hubei Province from 1949 to 2020.** Note: All time series have undergone Z-score standardization, so there are no original units. The dashed line in the figure represents the linear trend line.

## Discussion and conclusions

### Discussion

**Research extension and comparison.** Temporal Variations in Cholera Epidemics in Hubei Province. First, the 25-year cholera silent period (1955–1979) identified in this study constitutes one of the salient features of its spatiotemporal distribution pattern (Fig 3). This phenomenon was a phased outcome of the combined effects of robust administrative interventions and a relatively closed social system under specific historical conditions [42,52]. In the early years of the

People's Republic of China, the implementation of strategies such as preventive vaccination, transportation quarantine, and rapid blockade of epidemic foci, coupled with the household registration system and the prevailing international environment at that time, restricted population mobility. This significantly reduced the risk of pathogen introduction and dissemination, thereby sustaining the "zero-reporting" status throughout this period. Additionally, the epidemic situation across the country and in neighboring provinces also reached historic lows during the same period [42], supporting the authenticity of the "epidemic-free period" at the regional level rather than representing a data anomaly unique to Hubei. However, the resurgence of cholera in Hubei Province since the 1980s was primarily driven by a shift in the underlying epidemiological mechanisms, which can be attributed to four key factors. First, the exponential increase in population mobility brought about by the reform and opening-up policy created favorable conditions for the introduction of *Vibrio cholerae El Tor biotype*, a novel pathogenic strain. Second, the enhanced environmental survivability and higher infectivity of this strain enabled it to easily breach the prior static defense system based on isolation control and vaccination [53]. Third, in the 1980s, drinking water safety issues remained unresolved in many regions, and a considerable number of cholera cases were caused by water source contamination. Fourth, some areas exhibited strategic flaws in epidemic prevention and control practices, including overly extensive blockade of epidemic sites, irrational coverage of chemoprophylaxis, and a prominent tendency of "prioritizing treatment over prevention". Moreover, the emphasis on economic development at the expense of social benefits led to delays in epidemic reporting and prevention, triggering the spread of cholera outbreaks. Thus, the cholera-free period of 1955–1979 represented a fragile epidemic prevention equilibrium achieved through administrative means within a closed system. In contrast, the epidemic trends since the 1980s have revealed that, against an open socioeconomic backdrop, the mechanism of cholera emergence has been restructured into a complex interplay of population mobility amid globalization, the evolutionary adaptation of pathogens themselves, and regional local environmental factors. At the same time, considering the evolution of China's national cholera control strategy, its approach has historically relied on robust public health measures, including the "three controls" (water, sewage, and food management), a sensitive surveillance system, and timely outbreak containment [42]. Although cholera vaccines were used as an adjunct in certain regions during the mid-20th century, they never became a cornerstone of the national prevention and control framework. This further suggested that the fluctuations in incidence observed in Hubei—such as the resurgence in the 1980s—were more attributable to shifts in environmental and socioeconomic drivers, as well as potential adjustments in the intensity of core sanitation measures. Next, this study found that cholera epidemics peaked between August and October, coinciding with the warm and humid summer-autumn seasons in Hubei Province. Against this seasonal high-risk background, large-scale family and social gatherings were frequent, particularly around the Mid-Autumn Festival and National Day. Existing research has demonstrated that inadequate food hygiene supervision at such collective dining events can serve as critical nodes for foodborne cholera outbreaks [52,54]. The epidemic records collected in this study also contain explicit documentation of cases linked to mass gathering [55]. Therefore, although the temporal resolution of the data precluded the statistical isolation of the specific risk associated with the holiday days themselves, the intensive dining culture during the "two festivals" likely overlapped with the inherent climatic risk period, jointly shaping the epidemic peak pattern observed from August to October. In addition, this study identified three significant cholera epidemic cycles of 29, 19, and 8 years, which may reflect the coupling of multi-scale driving processes (Fig 5). The 29a cycle aligned with the epidemiological cycle of "introduction-epidemic-suppression" for major *Vibrio cholerae* biotypes (e.g., the *El Tor biotype* in the 1960s) and concurrent socio-economic transformations (e.g., the Reform and Opening-up period), suggesting it was a composite biosocial cycle shaped jointly by pathogen, host, and social factors. The 19a cycle corresponded closely to the activity period of low-frequency climate modes such as the Pacific Decadal Oscillation, potentially modulating regional climatic and hydrological backgrounds and setting an environmental rhythm at the decadal scale for epidemics. The 8a cycle was associated with interannual climate variability such as the El Niño-Southern Oscillation and the triggering frequency of flood disasters, reflecting the role of extreme weather events as direct "triggers" for outbreaks. Together, these cycles

constituted the comprehensive temporal characteristics of cholera epidemics under multiple mechanisms, including long-term pathogen evolution, medium-term climatic fluctuations, and short-term disaster triggers.

Regarding the spatial changes of cholera epidemics in Hubei Province. The shift in the epidemic's center of gravity was not merely a result of the simple accumulation of cases; instead, it reflected the transformation of core areas of population and socio-economic activities within the province (Fig 9f). During 1949–1959, the center was located in the Xiaogan region, which corresponded to the relatively higher population density and agricultural activity levels in the eastern Jianghan Plain (e.g., Xiaogan and Huangpi). With the rise of Wuhan as a central city in China after the 1980s, its massive population size and frequent internal and external exchanges made it an indisputable hotspot for cholera outbreaks in the 1980s-1990s, leading to a southward shift of the center to urban Wuhan (e.g., Jiangxia and Honghu). From the late 1990s to the 2000s, although the number of cases in Wuhan decreased, counties along the Yangtze River, such as Jingzhou and Jiangling, emerged as new key epidemic areas due to their role as secondary population centers in the province and their traditional reliance on water bodies for rice cultivation. This collectively contributed to a further southwestward shift of the center toward the hinterland of the Jianghan Plain (e.g., Honghu). This trajectory essentially outlined how the dynamic changes in the center of socio-economic activities and population distribution in Hubei Province over the past half-century had shaped the spatial pattern of infectious disease risks.

Regarding the influencing factors of cholera epidemics. First, methodologically, the SEM employed in this study offers a novel perspective beyond traditional association analyses for understanding the mechanisms driving regional cholera outbreaks. Most existing research relies on multiple regression or geoweight regression to identify risk factors. While these methods can reveal spatial heterogeneity, they struggle to quantify indirect effects and complex mediating pathways between factors. SEM's decomposition of causal pathways enables a more mechanistic understanding of how these factors influence cholera epidemics. Second, the results of the structural equation model (SEM) showed (Table 4) that nighttime light exhibited a significant positive total effect on cholera epidemic levels (1.080), yet this finding required cautious interpretation. Nighttime light is a typical high-order composite variable that is strongly correlated with both population density and GDP. It may simultaneously capture multiple dimensions of information, such as population agglomeration in urbanized areas, intensity of economic activities, density of infrastructure, and even the capacity for surveillance and reporting. The coexistence of its positive direct effect (1.139) and negative indirect effect (-0.059) suggested that there may be unfully identified collinear structures within the model, or inhibitory effects generated through other pathways (e.g., improved medical response capabilities). Therefore, nighttime light should not be simply interpreted as an independent "risk driver"; instead, it ought to be regarded as a comprehensive indicator of "risk-surveillance co-occurrence" that marked regional development levels. Furthermore, the study found that GDP and nighttime light exerted positive impacts on cholera epidemic levels, which contradicts the common understanding that "economic development improved sanitation facilities and reduced the risk of waterborne infectious diseases". Two potential reasons for this discrepancy were hypothesized: First, real risk amplification effect. Economically developed and highly urbanized regions such as Wuhan, the large population size, frequent cross-regional mobility, dense transportation networks, and overloaded sewage treatment systems collectively create a favorable environment for cholera transmission. Historically, multiple cluster outbreaks had been concentrated in such areas, indicating that socio-economic activities themselves might exert an "amplification effect" on epidemic spread. Second, surveillance and reporting bias. Economically developed regions were equipped with more robust infectious disease surveillance systems, advanced diagnostic capabilities, and stringent reporting protocols, resulting in far higher case detection and reporting rates than underdeveloped regions. The observed "high epidemic levels" might not stem entirely from differences in actual incidence; instead, they could be systematic errors caused by disparities in public health surveillance capacity—a problem that was difficult to avoid in retrospective studies utilizing passively reported data.

Regarding the comparison of this study's findings with other research. Fist, the driving mechanisms of cholera in Hubei Province revealed in this study not only align with the general patterns of global cholera epidemics but also exhibit distinct

characteristics due to environmental differences between inland water networks and coastal brackish water convergence zones. This study found that the average summer temperature was one of the strongest environmental factors driving cholera epidemics. This finding was highly consistent with observations from major global cholera-endemic regions. In Bangladesh, rising sea surface temperatures had been confirmed as a key driver of seasonal epidemics and interannual outbreaks; in the African Great Lakes region, temperature increases following the rainy season often coincided with epidemic outbreaks. Together, these findings corroborated the core role of temperature as a "universal switch" regulating the proliferation rate of *Vibrio cholerae* in aquatic environments [17,21]. However, the uniqueness of this case lay in its purely inland freshwater environment. Unlike coastal and estuarine epidemic areas (such as the Bay of Bengal), which were strongly influenced by salinity gradients and tidal effects, the river and lake networks in Hubei Province exhibited stable and extremely low salinity. Our model indicated that the direct impact of river network density itself was weaker, suggesting that in freshwater systems, dense water networks provided the necessary medium for transmission but are not a sufficient condition determining epidemic intensity. The spatial differentiation of epidemics was primarily determined by the degree of population aggregation and climatic suitability windows superimposed on this hydrological background. This "hydrological base + socio-climatic amplifier" mechanism provideed a theoretical framework for understanding cholera risks in other inland water-rich regions (such as large lake basins). Second, the transmission pathways of cholera were undergoing a global evolution. The increased records of outbreaks triggered by group gatherings in Hubei Province since the 1980s, as observed in this study, were not an isolated phenomenon. Instead, they reflected a global trend shifting from traditional "waterborne transmission" dominance toward an increasing importance of "foodborne transmission" [54,56]. As urban and rural water supply and sanitation infrastructure improve, the proportion of outbreaks directly caused by drinking water contamination had declined. However, the lengthening of food supply chains and the frequent occurrence of group dining activities driven by globalization and urbanization had created new pathways for *Vibrio cholerae* cross-contamination through food. The case of Hubei Province demonstrated that even in inland regions, socio-economic development and lifestyle changes were reshaping the transmission dynamics of infectious diseases. This necessitated that modern cholera prevention and control must transcend a singular environmental sanitation perspective and incorporate systematic supervision of food supply chain safety and collective dining hygiene. In addtion, since 1949, the impact of disaster factors on cholera epidemics had gradually weakened. The root cause of this transformation lay in the progress China had achieved since 1949 in improving medical and health standards, sustaining economic development, and maintaining social stability, which had laid a more solid foundation for cholera prevention and control [27].

**Policy recommendations.** To effectively prevent and respond to future cholera outbreaks, based on research into the spatio-temporal patterns and formation mechanisms of cholera epidemics in Hubei Province, the following Hubei-specific prevention and control recommendations are proposed. These offer particular reference value for regions with dense river and lake networks: (1) Establish a Safe Drinking Water Assurance System. As the most fundamental public health measure for cholera prevention and control, ensuring drinking water safety requires systematically enhancing water safety assurance capabilities beyond existing environmental monitoring. This includes improving water treatment and disinfection capacity to guarantee the stable effectiveness of chlorination and other processes. During early warning periods, temporarily increase residual chlorine levels in treated water before distribution and intensify testing of terminal pipelines in older urban areas. Addressing shortcomings in rural decentralized water supply, advance projects to consolidate and enhance drinking water safety. Promote small-scale water purification facilities and household disinfection equipment, while strengthening training for water management personnel and users on water quality assessment and equipment maintenance. (2) Hubei's unique breakfast culture of eating breakfast on the street carries the risk of contamination from blanched raw foods. It is recommended to promote the "Double-Non" certification (non-river-water dishwashing, non-raw/cold ingredients) in scenic spots such as Hubu Lane in Wuhan, and incorporate it into the evaluation criteria for Hubei's (Jingchu) dietary intangible cultural heritage. This certification system can be embedded within the market supervision department's existing "quantitative grading for food safety" dynamic evaluation,

serving as a bonus point or specialty certification for "transparent kitchens". Market reputation incentives can encourage voluntary participation by businesses, avoiding additional mandatory administrative burdens. (3) Risk management during key periods: Focus should be placed on the high-incidence summer and autumn seasons, especially August and September. Based on the main cycles of 55 years and 29 years for cholera in Hubei Province, preventive measures should be taken in advance in predicted peak years. (4) Establish a climate-adaptive early warning and response system. Given the critical driving role of climate conditions in this research model, future global warming will prolong summer heatwaves and increase the frequency of extreme precipitation events. This is highly likely to extend the"risk window"for cholera transmission and increase outbreak frequency. Therefore, the prevention and control system must be climate-adaptive. Seasonal risk outlooks based on climate forecasts could be issued each spring. During El Niño development phases, advance assessments of their potential impact on the following year's epidemic should be conducted. Flood disaster warnings should be directly linked to cholera emergency surveillance activation mechanisms, enabling enhanced monitoring of water bodies and diarrheal symptoms in key areas within 72 hours post-disaster. (5) Hierarchical spatial management and control: Level-1 risk zones (Wuhan urban area, 3 counties along the Yangtze River in Xianning): Implement sluice biosurveillance stations, install automatic sampling and testing equipment at the sluices of the Yangtze River/Han River. When Vibrio levels exceed standards, trigger the closure of ecological gates to block the invasion of polluted water into urban inland lakes. To reduce implementation costs, this system can be integrated with existing water quality monitoring stations and water conservancy hubs operated by water resources departments. By sharing station locations, power supplies, and communication networks, it achieves functional convergence of hydrological and pathogen monitoring, forming a "multi-functional integrated monitoring platform". Level-2 risk zones (other counties and cities in the Jianghan Plain): Incorporate water Vibrio concentration into the assessment system of the river chief system; suspend fishery and tourism projects in river sections that fail to meet standards. This strategy can be directly embedded into China's existing "River and Lake Chief System" governance framework. Implementation requires only adding assessment indicators without creating new administrative bodies, offering significant advantages of low administrative resistance and high synergistic benefits. Level-3 risk zones (western Hubei Province mountainous areas): Set up quarantine stations at inter-county transportation hubs (such as the Three Gorges Dam in Yichang) to conduct random inspections on freight vehicles from historical epidemic areas. Leveraging existing joint enforcement mechanisms between transportation and agriculture departments, deploy additional health quarantine personnel during high-risk summer and autumn periods. This creates a flexible prevention model combining routine patrols with peak-period monitoring, enabling low-cost, high-efficiency interception of imported risks. (6) Joint prevention and control at transportation hubs: at transportation hubs such as Wuhan Tianhe Airport and Hankou Railway Station, establish symptom monitoring mechanisms for individuals entering Hubei Province. For flights and trains arriving from cholera-endemic areas, implement active symptom-based screening. Conduct rapid diagnostic tests (RDTs) or PCR testing for Vibrio cholerae on stool samples from symptomatic individuals identified during screening, particularly those with diarrhea, and establish a "real-time case information sharing platform" with surrounding provinces to block the transmission chain of imported epidemics. (7) Ecological management and control of rivers and lakes: Establish "cholera prevention and control ecological corridors" in key river sections such as the Wuhan section of the Yangtze River and the Xiangyang section of the Han River. It is strictly prohibited to discharge domestic sewage into rivers during the flood season. Complete the construction of "regular monitoring points for *Vibrio cholerae*" at all drinking water sources at or above the county level by May each year, and implement closed-loop management of "supply suspension— disinfection—re-inspection" for water bodies that exceed standards. Furthermore, "ecological corridor" development should be deeply integrated with the national "Yangtze River Protection" strategy and local "Beautiful Rivers and Lakes" initiatives, translating pathogen control targets into specific ecological design parameters for wetland restoration and shoreline remediation projects. Monitoring station construction can be incorporated into routine fiscal safeguards for drinking water sanitation.

PLOS Neglected Tropical Diseases

**Research limitations.**

(1) Cholera Data. The cholera epidemic data employed in this study were derived from systematic collection of historical documents. However, their quality was inevitably constrained by the varying levels of development of surveillance systems across different periods and regions—a common challenge faced in all historical disease research. First, diachronic changes in surveillance sensitivity might affect trend interpretation. In the early period (1949–1979), when the surveillance system was in its initial stage of establishment, potential case underreporting suggested that the "low-incidence periods" or "disease-free periods" documented in this study might, to a certain extent, reflect the limited surveillance capacity of that era rather than the absolute eradication of the disease. In contrast, the improvement of the surveillance network and the standardization of reporting systems since the 1980s and 1990s might have led to a "statistical increase" in recorded case numbers, which was superimposed on the actual epidemic fluctuations. Second, the uneven spatial coverage of surveillance might have amplified the observed east-west disparities. In the early stages, medical resources, surveillance awareness, and reporting systems in Wuhan (the provincial capital) and counties in the eastern plains were superior to those in the remote mountainous areas of western Hubei Province. Therefore, the sparse or even zero recorded cases in western Hubei Province were likely the combined result of "genuine low disease risk" and "insufficient sensitivity of the surveillance system". This could have led our model to overestimate the protective effects of natural geographical factors (e.g., high elevation, low river network density) and to reinforce the "ldense in the east and sparse in the west" spatial pattern. While reflecting real disease risks, the observed pattern might also have, to some degree, exaggerated the actual disparities between eastern and western regions. Thus, simply classifying western Hubei Province as a "low-risk area" might lack precision; a more rigorous characterization was that it was a "low-reporting area confirmed based on existing data", and its true epidemic situation awaits further investigation. Despite these limitations, we had strived to construct the most complete historical epidemic sequence through systematic cross-validation of multi-source local chronicles and yearbooks. Future research can further refine the reconstruction of historical epidemic dynamics by mining finer-grained historical archives.

(2) Environmental factor data. The consideration of factors affecting cholera epidemics was not fully comprehensive. First, water and sanitation facilities (such as safe drinking water coverage and hygienic toilet prevalence) are critical components in interrupting cholera transmission. However, due to the study's long time span (1949–2020) and county-level spatial scale, it was impossible to obtain continuous, standardized quantitative data covering the entire period and all regions. Consequently, these factors could not be incorporated into the model. Second, Hubei Province's health-care standards (including county-level diagnostic and treatment capabilities and infectious disease prevention systems) represent variables influencing epidemic spread and control. However, pre-2000 data on county-level health conditions (such as rural sanitation facility coverage) and and medical standards (e.g., specialized diagnostic and treatment capabilities for intestinal infectious diseases) prior to 2000 lacked standardized statistical systems. Consequently, quantitative data suitable for long-term, county-level analysis were unavailable, precluding their inclusion in this quantitative model. Additionally, water pH—a critical environmental factor affecting Vibrio cholerae survival and reproduction—could not be analyzed due to the near-total absence of pH monitoring data for county-level rivers and lakes prior to 2000. Similarly, historical constraints limited the availability of sunshine duration data for Hubei Province during its early period (1949–1982), preventing its inclusion in the quantitative analysis of cholera outbreak mechanisms. To address these limitations, future research could supplement historical data on sanitation conditions, medical standards, water pH levels, and meteorological conditions through systematic review of local chronicles' sections on "sanitation" and "epidemic prevention", retrospective analysis of historical archives from environmental monitoring agencies (e.g., water quality records from hydrological stations), and enhanced mining and interpolation of historical meteorological data. Building upon this foundation, more comprehensive multi-factor quantitative analyses could be conducted to deepen understanding of the driving mechanisms behind cholera epidemics.

(3)  Structural Equation Modeling. The sample size for the structural equation model in this study comprised 103 county-level units. While meeting the basic requirements for model fitting, it remains relatively small compared to the generally recommended sample size (typically n > 200). This may result in insufficient statistical power for testing variables with weaker effects in the model (such as river network density), while estimates for variables with strong effects (such as population density and summer temperature) remain relatively stable. This limitation partially affects the stability and generalization ability of the model parameters. Future studies utilizing longer time series or higher spatial resolution data could enhance the model's robustness and interpretability. Additionally, the model inadequately accounts for temporal dynamics. The structural equation model employed a time aggregation strategy to identify steady-state drivers shaping long-term spatial patterns, thereby failing to capture dynamic characteristics during cholera epidemics. First, the model did not incorporate time-lag effects, such as the potential months- or even years-long delay between major floods and subsequent cholera outbreaks. Second, the current cross-sectional model did not address potential non-stationarity in the time series of individual variables and cholera itself, nor did it account for autocorrelation within county-level units over time. Although presented as exploratory analysis, Fig 11 displays standardized time series of key variables alongside cholera cases, visualizing their covariate trends and potential lagged relationships. However, rigorous validation of time lags, analysis of non-stationary series, and construction of spatio-temporal panel models require more complex econometric methods applied to more complete, higher-frequency panel data—representing a key direction for future research.

## Conclusions

In terms of time variation. The cholera epidemics in Hubei Province from 1949 to 2020 showed a pattern of "concentrated high-incidence periods, significant periodic fluctuations, and prominent seasonal characteristics". Its long-term downward trend and multi-scale fluctuation cycles essentially reflect the long-term interplay between climate stability, social prevention and control capabilities, and pathogen variation. The high incidence in summer and autumn highlights the key driving role of the warm and humid summer environment in the reproduction of *Vibrio cholerae*.

In terms of spatial distribution. Based on existing data, the cholera epidemics in Hubei Province exhibits a pattern of "dense in eastern Hubei Provinceand sparse in western Hubei Province", along with the characteristics of "agglomeration along rivers and lakes, and more severe in plains than mountains", reveals the decisive role of geographical environment and human activities in the spatial differentiation of the epidemic. The "westward and southward shift" of the epidemic's core reflects the reshaping of the epidemic's spread path by regional development disparities and the allocation of prevention and control resources.

In terms of formation mechanism. The temporal and spatial patterns of cholera epidemics in Hubei Province are the result of the combined effects of natural factors, disaster factors and human factors at multiple scales; Population density and summer temperatures were the direct factors driving the spread of cholera epidemics, while river networks formed the basic environmental background that facilitated transmission. Natural environmental factors such as drought and flood disasters, elevation, and precipitation, as well as human environmental factors including road networks and economic conditions, could not only directly drive the spread of epidemics but also regulate epidemics development through indirect pathways. Its formation mechanism provides a typical case for understanding the "environment-society-disease" interaction and offers a theoretical basis for formulating differentiated "regional and time-phased" prevention and control strategies.

## Supporting information

**S1 Data. Cholera Epidemics and Environmental Factors data in Hubei Province, China, 1949–2020.**
(XLS)

**S1 Table. Data Source Table of Cholera Epidemics.**
(XLS)

**S2 Table. Comparison Table of Adjustments to County-Level Administrative Divisions in Hubei Province (1949–2020).**
(XLS)

**S1 File. Matlab code for Mann-Kendall test.**
(TXT)

## Author contributions

**Conceptualization:** Zhiyu Chen, Shengsheng Gong, Tao Zhang.

**Data curation:** Zhiyu Chen, Tao Zhang.

**Formal analysis:** Zhiyu Chen, Tao Zhang.

**Funding acquisition:** Tao Zhang.

**Investigation:** Zhiyu Chen, Tao Zhang.

**Methodology:** Zhiyu Chen, Tao Zhang.

**Project administration:** Zhiyu Chen, Tao Zhang.

**Resources:** Zhiyu Chen, Tao Zhang.

**Software:** Zhiyu Chen, Tao Zhang.

**Supervision:** Shengsheng Gong, Tao Zhang.

**Validation:** Zhiyu Chen, Tao Zhang.

**Visualization:** Zhiyu Chen, Tao Zhang.

**Writing – original draft:** Zhiyu Chen.

**Writing – review & editing:** Zhiyu Chen, Shengsheng Gong, Tao Zhang.

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
