## [Decision Letter · Decision Letter 0]

13 Nov 2025

The Spatio-temporal Patterns and Formation Mechanisms of Cholera Epidemics in Hubei Province , China from 1949 to 2020

Dear Dr. Zhang,

Thank you for submitting your manuscript to PLOS Neglected Tropical Diseases. After careful consideration, we feel that it has merit but does not fully meet PLOS Neglected Tropical Diseases's publication criteria as it currently stands. Therefore, we invite you to submit a revised version of the manuscript that addresses the points raised during the review process.

Please submit your revised manuscript within by Jan 12 2026 11:59PM. If you will need more time than this to complete your revisions, please reply to this message or contact the journal office at plosntds@plos.org. Please include the following items when submitting your revised manuscript:

We look forward to receiving your revised manuscript.

Kind regards,

Mohammad Jokar, DVM

Guest Editor

Mathieu Picardeau

Section Editor

Shaden Kamhawi

co-Editor-in-Chief

Paul Brindley

co-Editor-in-Chief

**Journal Requirements:**

At this stage, the following Authors/Authors require contributions: Zhiyu Chen, Shengsheng Gong, and Tao Zhang. Please ensure that the full contributions of each author are acknowledged in the "Add/Edit/Remove Authors" section of our submission form.

4) Please upload a copy of Figures 10, and 11 which you refer to in your text on pages 16, and 21. Or, if the figure is no longer to be included as part of the submission please remove all reference to it within the text.

Potential Copyright Issues:

i) Figures 1, 8, and 9. Please (a) provide a direct link to the base layer of the map (i.e., the country or region border shape) and ensure this is also included in the figure legend; and (b) provide a link to the terms of use / license information for the base layer image or shapefile. We cannot publish proprietary or copyrighted maps (e.g. Google Maps, Mapquest) and the terms of use for your map base layer must be compatible with our CC BY 4.0 license.

6) We note that your Data Availability Statement is currently as follows: "All data underlying the findings described in this manuscript are publicly available without restriction.". Please confirm at this time whether or not your submission contains all raw data required to replicate the results of your study. Authors must share the “minimal data set” for their submission. PLOS defines the minimal data set to consist of the data required to replicate all study findings reported in the article, as well as related metadata and methods (https://journals.plos.org/plosone/s/data-availability#loc-minimal-data-set-definition).

**Reviewers' Comments:**

Reviewer's Responses to Questions

**Key Review Criteria Required for Acceptance?**

**Methods**

-Are the objectives of the study clearly articulated with a clear testable hypothesis stated?

-Is the study design appropriate to address the stated objectives?

-Is the population clearly described and appropriate for the hypothesis being tested?

-Is the sample size sufficient to ensure adequate power to address the hypothesis being tested?

-Were correct statistical analysis used to support conclusions?

-Are there concerns about ethical or regulatory requirements being met?

Reviewer #1: The manuscript is suitable for the journal's readership, as it addresses a relevant and timely topic related to the environmental and climatic determinants of cholera. The study provides valuable insights into the long-term relationship between cholera incidence and ecological, climatic, and demographic variables in Hubei Province, China, from 1949 to 2020. Using a combination of statistical and spatial correlation models, the authors effectively identify the significant factors associated with disease dynamics, including population density, temperature, precipitation, drought and flood frequency, as well as economic development. The manuscript makes a meaningful contribution to the literature on infectious disease ecology and environmental health.

One of the manuscript's main strengths lies in its use of multiple complementary methodologies, which allows for a robust and multidimensional interpretation of the data. The integration of climatological, hydrological, and socioeconomic information strengthens the analytical framework and enhances the validity of the findings. Additionally, the use of an extensive temporal dataset spanning more than seven decades provides a unique historical perspective on the evolution of cholera risk factors in the region. This longitudinal approach is particularly commendable, as it captures long-term environmental changes and their cumulative effects on disease patterns.

To further improve the clarity and contextual understanding of the manuscript, the authors are encouraged to define all acronyms upon first mention, ensuring accessibility for a broad readership that may include professionals from interdisciplinary backgrounds. Furthermore, to help readers situate the study within the broader public health context, it would be helpful to include a concise description of China's national cholera prevention program, including the year the vaccination program was introduced, the type of vaccine employed, and its estimated effectiveness. This contextual information would enable readers to understand better how policy interventions may have impacted cholera trends during the study period.

Regarding the limitations, although the authors acknowledge potential data constraints, it is recommended that the discussion be expanded to include underreporting, particularly in the early decades of the study period, and that the potential impact of this limitation on the results be addressed. The years following 1949 were characterized by limited surveillance capacity and inconsistent reporting practices, which could have led to an underestimation of case numbers and, consequently, affected the identification of long-term trends. A deeper reflection on this issue would enhance the transparency and critical depth of the discussion section.

In the recommendations, most of the proposed measures focus appropriately on environmental control strategies, such as integrating water monitoring into surveillance systems. However, it would strengthen the public health relevance of the conclusions to explicitly include the recommendation of ensuring access to safe and potable water for the population. Given that contaminated water remains one of the principal transmission routes for Vibrio cholerae, such an addition would align the recommendations with the global health framework for cholera elimination promoted by the World Health Organization.

Reviewer #2: (No Response)

Reviewer #3: The objectives of the study become apparent but are not as clearly articulated as they might be. The study design is mostly appropriate, with limitations likely beyond the control of the authors (due to requirement for historical data). I am not well placed to comment on all the statistical approaches used, but I did wonder if some approaches were being used because they could be (M-K test, Wavelet analysis), rather than because they needed to be. Examples (references) of where these analyses have been used for similar purposes would help justify the analytic approach used in this study.

Variables investigated seemed a bit haphazard, though all could potentially have some impact on V. cholerae. More justification for some variables (e.g. nighttime lighting) would help justify the study

**Results**

-Does the analysis presented match the analysis plan?

-Are the results clearly and completely presented?

-Are the figures (Tables, Images) of sufficient quality for clarity?

Reviewer #1: Please see above

Reviewer #2: (No Response)

Reviewer #3: Data presentation is very good, and results are clearly presented in the form of figures and tables. My question, as stated elsewhere, is are all analyses justified?

**Conclusions**

-Are the conclusions supported by the data presented?

-Are the limitations of analysis clearly described?

-Do the authors discuss how these data can be helpful to advance our understanding of the topic under study?

-Is public health relevance addressed?

Reviewer #1: Please see above

Reviewer #2: (No Response)

Reviewer #3: Many of the conclusions are supported by the data presented. Though the first stated finding (periodic fluctuations, etc.) is entirely self-evident without complicated data analyses.

Part of the Discussion addresses vaccination as a preventative measure, but vaccination coverage is not analysed in the study. See general comments section for more on this.

**Editorial and Data Presentation Modifications?**

Reviewer #1: Please see above

Reviewer #2: (No Response)

Reviewer #3: Both the Results and Discussion and Conclusion sections need to be edited. Conclusions more typically come after the Discussion. The conclusions shouldn't be a numbered list. The Discussion is also essentially a numbered list.

**Summary and General Comments**

Reviewer #1: Please see above

Reviewer #2: (No Response)

Reviewer #3: Cholera outbreaks have sporadically occurred in Hubei, and it is pleasing to see data analysis of those outbreaks. Such analyses can lead to more targeted interventions, as is the stated intention of the manuscript. I feel that some of the discussion points, while important in the control of cholera, were not directly relevant to the analyses conducted. I think the manuscript could be made shorter and more focused, and would be better for it. The authors must clearly state what the aims were, and then focus on those outcomes in the discussion.

As stated in the early section of the Discussion "The cholera outbreaks in Hubei during the 1980s were closely linked to the introduction of this new pathogen". Indeed, it is the presence of the pathogen that is likely the most important factor in whether a cholera outbreak occurs. I acknowledge that other factors are important (presence alone does not result in a cholera outbreak), so this study does add to the body of knowledge around cholera transmission in China, and indeed globally to some extent.

While the standard of writing is generally good, it is a long manuscript and should be edited to make it shorter and improve clarity. In doing so, please ensure you adhere to conventions of scientific writing, not the least italics for genus and species names!

Some minor points to improve clarity:

State approximate year range of Jiaqing-Daoguang period of the Qing Dynasty for benefit of international readers

“especially after the 21st century” we are currently in the 21 century, so change to “after 2000” or “during 21st century”

"The climate is mild and humid, with abundant rainfall, no severe cold in winter, and no cool weather in summer".

Add specific details such as annual rainfall and min and max average daily temperatures.

“Third, overly broad implementation of epidemic foci blockade and drug-based prevention, combined with insufficient human and material resources, resulted in ineffective intervention measures.” Please clarify this sentence.

Please provide more information about past vaccination programs in China.

“For instance, the prolonged absence of cholera outbreaks in Hubei Province after 1949 was closely associated with strict preventive vaccination measures.” To the best of my knowledge, oral cholera vaccines became available in the 1980s. So what were the strict preventative vaccination measures after 1949? If vaccination has been in place for a long time (maybe 30 – 40 years) perhaps state when it was first used in this region.

It is also implied that vaccination was being used in 1954? If so, as it a locally developed vaccine? Some more details are required. Reference 50 is presumably not widely available, and appears to not be a peer-reviewed article.

PLOS authors have the option to publish the peer review history of their article (what does this mean?If published, this will include your full peer review and any attached files.). If published, this will include your full peer review and any attached files.

**Do you want your identity to be public for this peer review?** For information about this choice, including consent withdrawal, please see our For information about this choice, including consent withdrawal, please see our Privacy Policy ..

Reviewer #1: No

Reviewer #2: No

Reviewer #3: **Yes:** Andrew GreenhillAndrew Greenhill

**Figure resubmission:**
---

## [Decision Letter · Decision Letter 1]

30 Jan 2026

Response to Reviewers '. This file does not need to include responses to any formatting updates and technical items listed in the 'Journal Requirements' section below. * A marked-up copy of your manuscript that highlights changes made to the original version. You should upload this as a separate file labeled 'Revised Manuscript with Track Changes '. * An unmarked version of your revised paper without tracked changes. You should upload this as a separate file labeled 'Manuscript '. If you would like to make changes to your financial disclosure, competing interests statement, or data availability statement, please make these updates within the submission form at the time of resubmission. Guidelines for resubmitting your figure files are available below the reviewer comments at the end of this letter. We look forward to receiving your revised manuscript. Kind regards, Mohammad Jokar, DVMGuest EditorPLOS Neglected Tropical Diseases Mathieu PicardeauSection EditorPLOS Neglected Tropical Diseases

Shaden Kamhawi

co-Editor-in-Chief

Paul Brindley

co-Editor-in-Chief

**Reviewers' comments:** Reviewer's Responses to Questions

**Key Review Criteria Required for Acceptance?**

**Methods**

-Are the objectives of the study clearly articulated with a clear testable hypothesis stated?

-Is the study design appropriate to address the stated objectives?

-Is the population clearly described and appropriate for the hypothesis being tested?

-Is the sample size sufficient to ensure adequate power to address the hypothesis being tested?

-Were correct statistical analysis used to support conclusions?

-Are there concerns about ethical or regulatory requirements being met?

Reviewer #1: The authors have made substantial improvements in response to the reviewers' comments. However, several issues still require clarification.

The explanation provided for river network density is appreciated. The authors state that, compared with the other two endogenous variables (population density and annual mean summer temperature), the effect value of river network density was relatively low. However, it would be essential to explicitly clarify that this variable was not statistically significant in the model.

There are some inconsistencies between the text and Table 4. For example, the reported mean elevation corresponds to the direct effect; the value for annual mean sunshine in the table is 0.110 (not 0.092); and the text describes the direct effect rather than the indirect effect. These discrepancies should be carefully reviewed and corrected to ensure consistency between the narrative and the table results.

The authors are also encouraged to use consistent terminology throughout the manuscript to refer to the same concept (for example, using either elevation or altitude, but not both interchangeably).

Finally, it appears the study's emphasis has shifted, as reflected in the conclusion that "population density and summer temperatures were the direct factors driving the spread of cholera epidemics, while river networks formed the basic environmental background that facilitated transmission." I recommend that the entire manuscript be carefully revised to ensure that the text is entirely consistent with this conclusion. The text would benefit from English-language editing.

Reviewer #2: (No Response)

Reviewer #4: 1) Method Section – Data Source Paragraph:

The text currently states: “we measure the severity of cholera epidemics using the number of cholera cases as an indicator.”

I suggest replacing “severity” with “intensity”, as severity is usually measured by deaths, case fatality rate (CFR), or mortality rates, whereas the number of cases reflects intensity.

2) Wavelet Analysis Periods:

I would have conducted the wavelet analysis for the third and fourth periods (1980–2020), given the length and uncertainty surrounding the "cholera-free period". Discussing a 29-year cycle seems less meaningful. It would be more informative to focus on recent cycles when the disease is more frequent and recurrent.

3) Mann-Kendall Test:

I find the M-K test not particularly relevant, as the graph already shows that while the frequency of outbreaks is increasing, the intensity appears to be decreasing.

4) SEM Indicators – Water and Sanitation:

Water and sanitation access were not included as indicators in the structural equation model (SEM). If including them is not feasible due to data unavailability for the study period or the scale of analysis (e.g., county level), I recommend explicitly noting this as a limitation of the study.

Reviewer #5: 1. The objectives are articulated with greater precision. However, the authors should clearly address how these objectives align with the existing gaps in the literature.

2. The study design is well-organized, using an integrated approach that combines both spatial and temporal analyses. The methodological rigor is also effectively aligned with the research questions posed, which enhances the overall validity and reliability of the manuscript. A clearer statement is needed to elaborate on the rationale behind the chosen methodology and how the research objectives provide insights into the design's robustness.

3. The authors provide a clear and detailed description of the population under study with the characterstic appropriate for the hypothesis being tested. Suggestions for Improvement such as further elaboration of any demographic variations within the population and their potential impact on the results.

4. The study's sample size is a significant strength, being sufficient, meeting convectional standards, and ensuring statistical power. Further discussion addressing how the sample size relates to different variables would add robustness to the conclusions.

5. The statistical analysis used is appropriate and effectively supports the conclusions drawn. To further strengthen the methodological credibility, it would help readers to see a clear rationale for why these particular tests were chosen over others. Including center of gravity analyses, seasonal distribution visuals, and local indicators of spatial autocorrelation would enhance interpretability. Besides, comparing their choice of Structural Equation Modeling (SEM) and its alignment with the conceptual framework to previous studies that have utilized similar frameworks could further strenthen the manuscript's analytical contribution.

6. The authors’ acknowledgment of ethical compliance is significant strength of the manuscript. Further justification on how informed consent and data confidentiality were managed in practice would boost the ethical rigor.

**Results**

-Does the analysis presented match the analysis plan?

-Are the results clearly and completely presented?

-Are the figures (Tables, Images) of sufficient quality for clarity?

Reviewer #1: Please see above.

Reviewer #2: (No Response)

Reviewer #4: 1) Remove the text: "After floods, a large number of deaths of humans and poultry pollute the water environment, providing conditions for the breeding of pathogenic bacteria. These contaminated water sources could reach other areas via floods, expanding the scope of transmission."

- Human deaths do not themselves “breed” bacteria. The bacteria are shed in stool of infected individuals; corpses from uninfected individuals are not a source of cholera transmission.

- Poultry are not carriers of V. cholerae, so mentioning poultry deaths as a source of cholera bacteria is misleading.

- The mechanism should focus on fecal contamination and bacterial proliferation in water, not human or animal corpses.

2) Consider documenting the rationale for increased cholera transmission following floods, based on the following:After floods, sewage and fecal matter can overflow from sanitation facilities and contaminate water sources, creating favorable conditions for the accelerated transmission of Vibrio cholerae.

3) In your text: “Droughts slowed down the transmission of Vibrio cholerae by reducing water-borne transmission routes. In general, flood disasters promoted the prevalence and transmission of cholera, while drought disasters inhibited its transmission”, I suggest adding supporting references.

I also recommend clarifying that, while drought may reduce cholera transmission in your specific context, it can increase both water-borne and interpersonal transmission. For example, a study in sub-Saharan Africa documented that cholera outbreaks were more likely under drought conditions, as limited access to safe water reduces hygiene and forces people to rely on unsafe surface water or gather closely at limited water points (https://pubmed.ncbi.nlm.nih.gov/40250229/).

While drought may lower incidence in your study area, these findings cannot be generalized to all settings. Finally, for cholera, it is more accurate to refer to incidence rather than prevalence.

Reviewer #5: 1. The analyses used in the results section perfectly align well with the pre-established analysis plan demonstrating systematic and rigorous analytical framework. Two targeted improvements would enhance the discussion

a) Explicitly referencing specific sections of the analysis plan when discussing the results.

b) Contrasting the findings with contemporary studies for greater depth.

2. The results are presented with clarity. The Wavelet Variance Plot for Cholera (Fig. 6) is particularly insightful for revealing the dominant temporal cycles in the outbreak data. To fully deliver on the manuscript's promise of analyzing spatio-temporal patterns, the connection between this temporal analysis and the spatial clusters (Fig. 8) should be strengthened.

3. The figures and tables included in the manuscript are of a generally high quality and contribute effectively to the clarity of the results. Fig. 11 could benefit from additional annotations or emphasizing specific data points to draw attention to key findings.

**Conclusions**

-Are the conclusions supported by the data presented?

-Are the limitations of analysis clearly described?

-Do the authors discuss how these data can be helpful to advance our understanding of the topic under study?

-Is public health relevance addressed?

Reviewer #1: Please see above.

Reviewer #2: (No Response)

Reviewer #4: 1) This explanation is tempting and is one of my favorite part: “Thus, the cholera-free period of 1955–1979 represented a fragile epidemic prevention equilibrium achieved through administrative means within a closed system. In contrast, the epidemic trends since the 1980s have revealed that, against an open socioeconomic backdrop, the mechanism of cholera emergence has been restructured into a complex interplay of population mobility amid globalization, the evolutionary adaptation of pathogens themselves, and regional local environmental factors.” However, I would need a bit more to be convinced. Have you looked at cholera incidence in China during this period, or in neighboring provinces, to make sure this isn’t just a reporting artifact? If cholera was largely confined to coastal areas, that could explain why it reemerged in the 1980s, when population movement increased, such as seasonal workers traveling to coastal regions. Showing some of these data would strengthen your argument.

2) In the policy section, please clarify whether “on-arrival testing” refers to stool sample testing or water sample testing. To my knowledge, there is currently no licenced RDT available to detect Vibrio cholerae in water samples. Moreover, the GTFCC surveillance guidelines recommend testing stool samples rather than water samples. For feasibility and alignment with global guidelines, consider replacing “water samples” with “stool samples” in the phrase: “symptom monitoring for personnel entering Hubei Province and rapid testing of water samples in luggage.”

Additionally, you could consider implementing “on-arrival testing” for flights and trains arriving from cholera-affected areas, and establishing a “real-time sharing platform” with neighboring provinces to interrupt the transmission chain of imported cases.

3) Consider moving policy recommendations 6 and 7 on water access and food safety higher. Since drinking water safety is the cornerstone of cholera prevention, these measures deserve greater emphasis.

Reviewer #5: 1. The conclusions drawn in the manuscript are significantly supported by the data presented throughout the study.

2. The limitations of the analysis are addressed, although they could benefit from more detailed discussion of these limitations such as any restrictions arising from sample size, data availability, or methodological choices.

3. The authors effectively discuss how their findings advance the understanding of cholera in Hubei Province. To further strengthen the manuscript, they should also articulate the broader significance of their data for ongoing global cholera research and public health strategies.

4. The manuscript successfully highlights the public health importance of the cholera findings.

**Editorial and Data Presentation Modifications?**

Reviewer #1: Please see above.

Reviewer #2: (No Response)

Reviewer #4: Result section: The positive total effect for population density is 0.837 and not 0.807 in the text: "(1) Population density significantly increased the level of cholera epidemics and was the

most dominant endogenous factor contributing to cholera epidemics, exhibiting a positive

total effect (0.807)".

There is a duplication in the text:"The annual annual mean summer temperature in Hubei Province is

approximately 27°C, which is close to the temperature suitable for the reproduction of Vibrio

cholerae".

Reviewer #5: (No Response)

**Summary and General Comments**

Reviewer #1: Please see above.

Reviewer #2: (No Response)

Reviewer #4: The long study period, including the particularly insightful silent period and the exploration of factors that may explain it, is highly valuable. Congratulations for that! However, I would suggest focusing the statistical analyses (Mann-Kendall, Wavelet, and SEM) on the most recent 20-year period, while retaining descriptive analyses (which you have presented very well) for the entire 1949–2020 timeframe.

Reviewer #5: Overall Comments

The manuscript provides a substantial investigation into cholera epidemics in Hubei Province, China, offering valuable insights for epidemiology. It employs rigorous methodologies and has practical implications for public health interventions, though several areas need improvement to enhance clarity and impact.

Strengths

Clear Objectives: The study sets well-defined objectives that clarify its significance in understanding cholera patterns.

Robust Methodology: The integration of spatial and temporal analyses allows for a nuanced examination of cholera outbreaks, enhancing validity.

Public Health Relevance: The research emphasizes important implications for policymakers and health practitioners, particularly regarding intervention strategies.

Effective Data Presentation: Results are clearly presented with appropriate figures and tables, aiding reader comprehension.

Weaknesses

Limited Contextualization: There is insufficient comparison with existing literature, which could strengthen the significance of the findings.

Lack of Discussion on Limitations: A more thorough exploration of potential biases and methodological constraints is needed for a balanced perspective.

Ethical Considerations: The manuscript should provide a deeper discussion of ethical standards adhered to, particularly regarding informed consent and data confidentiality.

The study’s unique contribution lies in its comprehensive analysis, offering fresh insights into cholera transmission dynamics critical for effective public health responses.

Additional Minor Comments for the Authors

1. Data Reliability and True Absence of Cases (25 Years): The authors have addressed data reliability effectively by providing a more detailed discussion of data collection methods. However, they should further elaborate on the implications of any potential gaps in case reporting over the past 25 years. Including a more comprehensive description of validation processes would strengthen the credibility of the findings.

2. Variable Selection and Conceptual Framework of Structural Equation Modeling (SEM): The authors have made commendable progress in clarifying aspects of variable selection. Still, they should explicitly align their choices with the conceptual framework of SEM. Additionally, including examples of previous studies that have utilized similar frameworks could enhance the paper’s analytical contribution.

3. Statistical Rigor and Interpretation of Temporal Patterns: While the justification for statistical methods has improved, the authors should also consider providing comparisons with recent methodologies to further substantiate their claims, thereby enriching the discussion on statistical rigor.

4. Spatial Analysis and Presentation: The authors have made strides in improving spatial analysis by incorporating more visual representations. However, they should also consider adding center of gravity analyses, seasonal distribution visuals, and local indicators of spatial autocorrelation. Including these elements would enhance interpretability and provide deeper insights into the patterns observed.

5. The introduction is somewhat longer than necessary. Consider reducing it by approximately 20–30\%. The historical context on cholera epidemics and theoretical background can be condensed to allow the motivation and unique contributions of this study to emerge more clearly.

PLOS authors have the option to publish the peer review history of their article (what does this mean?If published, this will include your full peer review and any attached files.). If published, this will include your full peer review and any attached files.

**Do you want your identity to be public for this peer review?** For information about this choice, including consent withdrawal, please see our For information about this choice, including consent withdrawal, please see our Privacy Policy ..

Reviewer #1: No

Reviewer #2: No

Reviewer #4: No

Reviewer #5: No

**Figure resubmission:** While revising your submission, we strongly recommend that you use PLOS’s NAAS tool (https://ngplosjournals.pagemajik.ai/artanalysis) to test your figure files. NAAS can convert your figure files to the TIFF file type and meet basic requirements (such as print size, resolution), or provide you with a report on issues that do not meet our requirements and that NAAS cannot fix.

**Reproducibility:** To enhance the reproducibility of your results, we recommend that authors of applicable studies deposit laboratory protocols in protocols.io, where a protocol can be assigned its own identifier (DOI) such that it can be cited independently in the future. Additionally, PLOS ONE offers an option to publish peer-reviewed clinical study protocols. Read more information on sharing protocols at https://plos.org/protocols?utm_medium=editorial-email&utm_source=authorletters&utm_campaign=protocols To enhance the reproducibility of your results, we recommend that authors of applicable studies deposit laboratory protocols in protocols.io, where a protocol can be assigned its own identifier (DOI) such that it can be cited independently in the future. Additionally, PLOS ONE offers an option to publish peer-reviewed clinical study protocols. Read more information on sharing protocols at https://plos.org/protocols?utm_medium=editorial-email&utm_source=authorletters&utm_campaign=protocols

---

## [Decision Letter · Decision Letter 2]

8 Mar 2026

Dear Dr. Zhang,

We are pleased to inform you that your manuscript 'The Spatio-temporal Patterns and Formation Mechanisms of Cholera Epidemics in Hubei Province , China from 1949 to 2020' has been provisionally accepted for publication in PLOS Neglected Tropical Diseases.

Best regards,

Mohammad Jokar, DVM

Guest Editor

Mathieu Picardeau

Section Editor

Shaden Kamhawi

co-Editor-in-Chief

Paul Brindley

co-Editor-in-Chief

Reviewer's Responses to Questions

**Key Review Criteria Required for Acceptance?**

**Methods**

-Are the objectives of the study clearly articulated with a clear testable hypothesis stated?

-Is the study design appropriate to address the stated objectives?

-Is the population clearly described and appropriate for the hypothesis being tested?

-Is the sample size sufficient to ensure adequate power to address the hypothesis being tested?

-Were correct statistical analysis used to support conclusions?

-Are there concerns about ethical or regulatory requirements being met?

Reviewer #2: (No Response)

Reviewer #5: Yes, all comments haven addressed by the authors

**Results**

-Does the analysis presented match the analysis plan?

-Are the results clearly and completely presented?

-Are the figures (Tables, Images) of sufficient quality for clarity?

Reviewer #2: (No Response)

Reviewer #5: Yes, all comments haven addressed by the authors

**Conclusions**

-Are the conclusions supported by the data presented?

-Are the limitations of analysis clearly described?

-Do the authors discuss how these data can be helpful to advance our understanding of the topic under study?

-Is public health relevance addressed?

Reviewer #2: (No Response)

Reviewer #5: Yes

**Editorial and Data Presentation Modifications?**

Reviewer #2: (No Response)

Reviewer #5: (No Response)

**Summary and General Comments**

Reviewer #2: (No Response)

Reviewer #5: (No Response)

PLOS authors have the option to publish the peer review history of their article (what does this mean?If published, this will include your full peer review and any attached files.). If published, this will include your full peer review and any attached files.

**Do you want your identity to be public for this peer review?** For information about this choice, including consent withdrawal, please see our For information about this choice, including consent withdrawal, please see our Privacy Policy ..

Reviewer #2: No

Reviewer #5: No

---

## [Editor Report · Acceptance letter]

Dear Dr. Zhang,

We are delighted to inform you that your manuscript, "The Spatio-temporal Patterns and Formation Mechanisms of Cholera Epidemics in Hubei Province , China from 1949 to 2020," has been formally accepted for publication in PLOS Neglected Tropical Diseases.

Best regards,

Shaden Kamhawi

co-Editor-in-Chief

Paul Brindley

co-Editor-in-Chief
